

# Cost of holographic path integrals

**Ammanamanchi Ramesh Chandra[1⋆], Jan de Boer[1†], Mario Flory[2‡],
Michal P. Heller[3∘], Sergio Hörtner[4§] and Andrew Rolph[1¶]**

**1** Institute for Theoretical Physics, University of Amsterdam,
1090 GL Amsterdam, The Netherlands
**2** Instituto de Física Teórica UAM-CSIC, 28049, Madrid, Spain
**3** Department of Physics and Astronomy, Ghent University, 9000 Ghent, Belgium
**4** Max Planck Institute for Gravitational Physics (Albert Einstein Institute),
14476 Potsdam-Golm, Germany

⋆ s.r.c.ammanamanchi@uva.nl , † j.deboer@uva.nl , ‡ mario.flory@csic.es ,
∘ michal.p.heller@ugent.be , § sergio.hoertner@aei.mpg.de , ¶ a.d.rolph@uva.nl

## Abstract

We consider proposals for the cost of holographic path integrals. Gravitational path integrals within finite radial cutoff surfaces have a precise map to path integrals in $T\overline{T}$ deformed holographic CFTs. In Nielsen's geometric formulation cost is the length of a not-necessarily-geodesic path in a metric space of operators. Our cost proposals differ from holographic state complexity proposals in that (1) the boundary dual is cost, a quantity that can be 'optimised' to state complexity, (2) the set of proposals is large: all functions on all bulk subregions of any co-dimension which satisfy the physical properties of cost, and (3) the proposals are by construction UV-finite. The optimal path integral that prepares a given state is that with minimal cost, and cost proposals which reduce to the CV and CV2.0 complexity conjectures when the path integral is optimised are found, while bounded cost proposals based on gravitational action are not found. Related to our analysis of gravitational action-based proposals, we study bulk hypersurfaces with a constant intrinsic curvature of a specific value and give a Lorentzian version of the Gauss-Bonnet theorem valid in the presence of conical singularities.



# 1 Introduction

A central question within AdS/CFT [1–3] is how the bulk geometry is encoded in the boundary. Developments over the last fifteen years have shown that entanglement in the boundary state plays a central role [4], and the Ryu-Takayanagi formula that calculates entanglement entropy holographically from areas of bulk surfaces has been a particularly useful tool in elucidating this connection [5]. However, entanglement entropy is not enough [6], with one justification for this claim being the existence of bulk geometries with subregions that Ryu-Takayanagi surfaces cannot probe [7,8].

The complexity of state preparation has been proposed as a new entry in the holographic dictionary, in a search for the boundary dual of the growth of black hole interior volume that persists well beyond the thermalisation time scale of conventional holographic probes such as the aforementioned entanglement entropy and simple correlation functions [6,9]. This complexity can been heuristically thought of, along the lines of Multiscale Entanglement Renormalization Ansatz (MERA) [10], as the size of a tensor network preparing a desired state in an underlying discretized conformal field theory starting from a disentangled initial state and using a minimal number of tensors. See [11] for a recent review of these developments.

The microscopic definition of state complexity is ambiguous because it requires a number of arbitrary choices: a set of elementary gates, a reference state, penalty factors and a margin of error. Furthermore, defining complexity in continuum QFTs is difficult; see [12, 13] and [14–17] for progress in free field theories and $1 + 1$-dimensional CFTs respectively. Nielsen's geometric approach to state complexity as the shortest path in a metric space of unitary operators is a promising approach [18], yet it faces challenges such as the identification of simple reference states in the continuum, the incorporation of mixed states and the regularization of UV divergences.

There are numerous proposed bulk duals to state complexity. The original complexity=volume (CV) proposal [19] relates complexity to the volume of a codimension-one maximal bulk slice anchored on the boundary at the relevant time. Other well-studied prescriptions for holographic complexity consider alternative characterizations of the black hole size: the complexity=action (CA) [20] and complexity=volume 2.0 (CV2.0) [21] proposals respectively relate state complexity to the on-shell gravitational action and volume evaluated on the Wheeler-DeWitt (WDW) patch, which is a codimension-zero bulk region with null boundaries. Recently it has been shown that there are actually infinitely many bulk quantities that exhibit the late-time linear growth and switchback effect which are characteristic of black holes [22]. The holographic complexity proposals have various arguments and justifications for why their bulk dual is the correct quantity for counting gates. For example, one can motivate the CV proposal by the observation that the MERA tensor network resembles a discretized version of the hyperbolic disk [23]. The Ryu-Takayanagi formula has a fairly rigorous derivation [24]; in contrast there is no compelling derivation of any holographic state complexity proposal.

The path-integral optimisation program, initiated in [25–27], is an interesting approach to complexity in field theory, and is part of the inspiration for the present paper. For recent developments see [28–31]. In this approach the preparation of a state by a Euclidean path integral is 'optimised' by, in a sense, coarse-graining the background metric. The physical idea is that while coarse-graining the background metric does change the UV structure of the prepared state, these UV differences are exponentially suppressed by the Euclidean time evolution. The least complex or optimum Euclidean path integral is that which coarse-grains the background metric as much as possible without affecting the prepared state below the UV cutoff. To coarse-grain the metric one performs a Weyl rescaling, which in 2d CFTs leaves the ground state wavefunctional invariant up to a Liouville action prefactor arising from the conformal anomaly. The Liouville action also appears in the next-to-leading order expansion in the UV cutoff of a proper, fine-tuned cost function for a particular quantum circuit constructed out of the energy-momentum tensor components of a two-dimensional CFT in Minkowski space [32–34]. This gives a Nielsen geometric interpretation to the Liouville action appearing in path integral optimisation of 2d CFTs, but this has not been generalised beyond 2d CFTs.

In a previous article we attempted to combine the ideas of path integral optimisation and holographic $T\overline{T}$, and to propose the cost of a bulk spacetime circuit which can be minimised to match state complexity [35].[1] More precisely, we suggested a quantum circuit interpretation

---

[1]Earlier work combining the idea of path-integral optimization and state complexity with $T\overline{T}$-deformations includes [36,37].

of Euclidean path integrals on codimension-0 bulk subregions bounded by two time slices and a radial cutoff surface, and proposed the cost to be given by the on-shell gravitational action of the subregion. The radial cutoff surface is related to an effective finite cutoff in the dual field theory in a way made precise by holographic $T\overline{T}$ [38]. By allowing that radial cutoff to vary we can coarse-grain the path integral in just the same way as in path integral optimisation. We showed that the radial cutoff surface that minimises the gravitational action when the initial state is trivial lands on the time slice on which the final state lives, and that the minimal cost equals the volume of that slice, which is in accordance with the CV holographic state complexity proposal. Despite this success several aspects of the ideas put forward in [35] remained unclear or unsatisfactory. Of all functions on bulk subregions it was not clear whether or why the gravitational action itself should play a distinguished role. Furthermore, the extension of the proposal to Lorentzian signature appeared to be problematic.

In this paper we initiate a more general and thorough analysis of possible notions of cost associated to holographic path integrals. The cost proposals we consider are of the general form

$$\mathscr{C} = f(X_M). \tag{1}$$

$\mathscr{C}$ is cost and $f(X_M)$ a function on bulk subregion $X_M$. $M$ is the boundary submanifold on which the path integral is performed, and to be meticulous we will give the precise map between bulk and boundary path integrals, including possible $T\overline{T}$ deformations if the bulk path integral is within a finite cutoff surface. The two parts of a given cost proposal that need to be specified are the bulk subregion $X_M$ that $M$ maps to, and the function $f$ on that subregion. $X_M$ may be a codimension-one bulk radial cutoff surface, it may be a codimension-0 region bounded by that radial cutoff surface and two covariantly defined Cauchy slices, or something more general.

Cost obeys a set of physical requirements that we use to substantially reduce the set of allowed cost proposals (1). For example, since cost is heuristically the number of gates in a quantum circuit it should be non-negative as well as additive under concatenation of quantum circuits. We find important differences in what are physically reasonable cost proposals between Lorentzian and the Euclidean path integrals.

It is arguably more natural to associate cost rather than complexity to functions on bulk subregions. One argument is that functions such as volume and gravitational action are integrals of local densities and so are additive on unions of disjoint bulk subregions. Complexity, as geodesic length in a metric space of operators, is not additive but subadditive. Another argument, for subregions with two boundaries that are partial Cauchy slices of Lorentzian bulks, is that the natural operator to associate to such bulk subregions is the path integral on that subregion, which builds up the operator in a series of infinitesimal Hamiltonian time evolutions. Physically there is no reason to suppose that this is the least complex way to build up the operator.

We can also argue for why functions on codimension-0 bulk subregions should be associated with operator cost rather than state cost. The time evolution operator acting on a CFT vacuum state does nothing. It traces a trivial empty path in the space of states, even though the operator is nontrivial, so we expect zero state cost but non-zero operator cost associated to this evolution. The time evolution of the CFT vacuum state is dual to the time evolution between two constant time slices of a pure AdS bulk, a codimension-0 subregion. A generic function such as volume or gravitational action evaluated on such a region will be non-zero, which rules it out as state cost, i.e. the length of a path between initial and final state.

We connect our path integral cost proposals to holographic state complexity proposals using ideas from path integral optimisation. We fix a bulk state and optimise over all possible holographic path integral preparations of that state. To do this we choose a cost proposal (1) and minimise cost over path integrals that prepare the same state. Path integrals on manifolds

with two boundaries define operators, and path integrals generate paths through the space of operators which are not generally geodesic, and so path integral cost upper bounds the operator complexity because that is the length of the shortest path to the operator. Operator complexity itself upper bounds state complexity, because state complexity can be defined as operator complexity minimised over those unitary operators that map a fixed reference state to the target state:

$$C(|f\rangle) := \min_{\{U:\, U|i\rangle=|f\rangle\}} C(U). \tag{2}$$

This optimisation procedure produces candidate notions of state complexity. It should be pointed out that the optimisation is restricted to holographic path integral preparations, so the relevant notions of complexity roughly correspond to gate sets which admit a semiclassical dual gravitational representation. The requirement that a state complexity proposal should be obtainable by optimising a suitable cost functional over gravitational path integral representations substantially restricts possible state complexity proposals. The complexity=volume and complexity=volume 2.0 proposals seem connected to the optimisation of suitable cost functionals, but we have not been able to find such a connection for the complexity=action proposal.

For state complexity proposals one often imposes linear late time growth in black hole backgrounds and the so-called switchback effect. While we make some preliminary remarks about these issues in section 4, we have not systematically analyzed which cost proposals agree with these requirements after optimisation and it would be interesting to study this in more detail. It is worth pointing out that in [22] a detailed study of possible codimension one covariant notions of state complexity was given and it was found that many of these are compatible with late time linear growth and the switchback effect, so perhaps imposing these extra criteria is not too restrictive. While [22] also considers general covariant functionals just as we do, it only considers state complexity and does not look at the more general holographic cost proposals, nor does it address this issue which of the former are the result of optimizing the latter, which is one of the key issues we address.

**Outline**

In section 2 we introduce the set of bulk path integrals we would like to think of as quantum circuits and associate a cost to. We review holographic $T\overline{T}$ and give the precise map from our bulk path integrals to boundary path integrals in $T\overline{T}$ deformed theories. We describe how to coarse-grain both the bulk and boundary theories to EFTs in which we expect UV-finite path integral cost. We explain in what sense we can 'optimise' the path integral between bulk states, and how to prepare a given bulk state in the least complex way.

In section 3 we map out the set of holographic proposals for path integral cost. Defining a holographic cost proposal requires specifying a bulk subregion and a function on that subregion. We give a set of physical requirements that any such cost proposal must satisfy: non-negativity, additivity, covariant definition, invariance under time reversal, and that the cost of the trivial path integral vanishes. We then describe how these requirements reduce the original space of holographic path integral cost proposals. The non-negativity requirement is particularly important; we point out that cost proposals using the Einstein-Hilbert gravitational action are generically unbounded from below.

In section 4 we connect our holographic path integral cost proposals to existing state complexity proposals. Specifically, we construct a set of path integrals that prepare a fixed bulk state and give two cost proposals that when minimised over this set of path integrals reduce to the CV state complexity proposal, and one which reduces to CV2.0. We are not able to find a cost proposal that reduces to the CA complexity proposal, and we discuss the difficulties in doing so. Finally we introduce a new candidate holographic complexity proposal constructed

Table 1: Summary of the subset of path integral cost proposals that are considered in detail in this paper. Includes the metric signature of the bulk to which the proposal applies to, which of the physical properties of cost that are satisfied, and, when all properties are satisfied, which holographic state complexity proposal the cost proposal reduces to.

| Cost equals... | Bulk signature | Satisfies physical properties of cost? | | | | | Reduces to which state complexity proposal? |
| --- | --- | --- | --- | --- | --- | --- | --- |
| | | Zero cost ⇔ trivial path integral | Additivity | Symmetry | Covariance | Non-negativity | |
| Codim-1 boundary volume | Euclidean | ✓ | ✓ | ✓ | ✓ | ✓ | CV[0] |
| Codim-1 boundary volume | Lorentzian | ✗ | ✓ | ✓ | ✓ | ✓ | N/A |
| Codim-0 bulk volume | Euclidean | ✓[1] | ✓ | ✓ | ✓ | ✓ | CV[0] |
| Codim-0 bulk volume | Lorentzian | ✓[1] | ✓ | ✓ | ✓ | ✓ | CV2.0[3] |
| Codim-0 gravitational action | Euclidean | ✓[1] | ✓ | ✓ | ✓ | ✓[2] | CV[0,4] |
| Codim-0 gravitational action | Lorentzian | ✓[1] | ✓ | ✓ | ✓ | ✗ | N/A |

[0] For time reflection-symmetric surfaces; see section 4.1.
[1] If initial and final bulk slices suitably defined; see section 4.
[2] Except for a few fringe cases; see section 3.4.
[3] Actually reduces to the volume of half a WdW patch; see section 4.2.
[4] Only explicitly shown for pure AdS; see [35] and section 4.3.

from constant scalar curvature surfaces, inspired by [35], and explore its time-dependent properties in the BTZ black hole setup. See table 1 for a summary of path integral cost proposals we found that satisfy the physical requirements and reduce to state complexity proposals.

In section 5 we reconsider the flow equation derived in [35] that describes the motion of a cutoff surface in a fixed Euclidean AdS background. We provide a particularly simple ansatz that in the three dimensional case captures generically the solutions of the flow equation. The solutions, which are interpreted as bounding regions of the bulk where the action does not change under infinitesimal displacements, turn out to be foliated by geodesics in the ambient AdS$_3$ manifold, and we provide examples thereof. We also show that in the Lorentzian case these solutions, which we then dub "lemons", provide an interesting and, to the best of our knowledge, novel foliation of the WDW patch.

We end the paper with a discussion of the challenges that the cost function approach to holographic complexity poses and point out possible lines of future work. Some technical discussions are relegated to appendices.

# 2 Path integrals between geometric states

How complex are states in semiclassical gravity? What is the least complex way of evolving from one state to another via a path integral? To address these questions we will first discuss how a given semiclassical bulk state in AdS gravity can be the solution to not one but a continuous family of mixed boundary conditions at the asymptotic boundary, and so have representations in many deformed holographic CFTs. Next, since this is a fine-grained description and the costs and complexities of these states are UV divergent, we will describe how precisely to coarse-grain the holographic theories to get UV-finite results. Lastly we explain in what sense we can 'optimise' the path integral between bulk states.

We will momentarily give a more precise description for the case of pure $AdS_3$ gravity, but first give a more heuristic picture of the general situation. Given a semiclassical bulk configuration which is asympotically AdS, it is a priori not yet clear in what theory this configuration describes a state, as we have not yet specified the boundary conditions. In other words, we have not yet defined which degrees of freedom fluctuate and which ones are kept fixed (i.e. what are the sources and what are the expectation values of operators), or equivalently, we have only provided one bulk configuration rather than the full phase space of solutions.

The standard choice would be to impose standard asymptotic AdS boundary conditions where the bulk configuration would correspond to a state in the dual CFT. We can however also make other interesting choices. For the purpose of the paper, we will be interested in imposing Dirichlet boundary conditions on a timelike hypersurface in the bulk (for Lorentzian space-times). Linearized on-shell fluctuations around this background which preserve the Dirichlet boundary conditions will have a mix of non-normalizable and normalizable modes turned on near the AdS boundary, which should therefore be interpreted as belonging to a deformed CFT with sources for multi-trace operators turned on. At the linearized level, one only finds double-trace deformations, and if the backreaction of the matter fields on the geometry can be neglected one only finds a double-trace deformation for the stress-tensor which looks like a $T\overline{T}$ deformation with a space-time dependent source. Including the backreaction of matter will generate other double-trace deformations which involve other operators in the CFT. Going beyond the linearized approximation, one will generically encounter higher-trace deformations as well. A more precise analysis would consider the full non-linear phase space of solutions with the relevant Dirichlet boundary conditions, but we do not expect this phase space to have simple asymptotics at infinity, as at the non-linear level the irrelevant higher-trace deformations will generally lead to solutions which are not asymptotically AdS. This full non-linear analysis is in general intractable, but luckily the situation in pure $AdS_3$ is more favorable and we can make some of these statements more precise as we will do next. The reader should keep in mind though that ultimately we are interested in the more general situation sketched here.

## 2.1 Review: holographic $T\overline{T}$

We consider path integrals bulk theories which are holographically dual to $T\overline{T}$ deformed CFTs. We start with a review of holographic $T\overline{T}$ in order to understand the precise holographic map between bulk and boundary path integrals. First we follow the perspective and presentation of [39], that $T\overline{T}$-deformed holographic CFTs are UV-complete but non-local field theories, and that they are dual to gravity in asymptotically AdS spacetime, i.e. whose bulk slices have infinite volume, with mixed boundary conditions at infinity. In the next subsection we discuss the coarse-grained descriptions of both sides: gravity with Dirichlet boundary conditions inside a finite cutoff surface, and the $T\overline{T}$ low energy effective theory.

### 2.1.1 Fine-grained description

For concreteness and simplicity we consider pure three dimensional semiclassical Einstein gravity in AdS with all bulk matter background fields in their vacuum configurations. In this case the holographic $T\overline{T}$ deformation, i.e. the deformation which brings in the bulk Dirichlet boundary conditions to finite cutoff, is the original $T\overline{T}$ deformation as studied by Zamolodchikov [38, 40]. As alluded to above, our discussion generalises to other dimensions and with non-trivial bulk background fields turned on with only a modification to the boundary field theory deformation [41–44]. The most general asymptotically AdS$_3$ metric solving Einstein's equations can be written in Fefferman-Graham gauge as

$$ds^2 = G_{\mu\nu}dx^\mu dx^\nu = \frac{L^2}{4\rho^2}d\rho^2 + G_{ij}(\rho,x)dx^i dx^j, \quad G_{ij}(\rho,x) = \frac{G_{ij}^{(0)}(x)}{\rho} + G_{ij}^{(2)}(x) + \rho G_{ij}^{(4)}(x). \quad (3)$$

Here, $\rho$ is a radial coordinate with the asymptotic boundary at $\rho = 0$.[2] For AdS$_3$, the Fefferman-Graham expansion ends at $G^{(4)}$, and on-shell $G^{(4)}$ is fully determined by $G^{(2)}$ and $G^{(0)}$:

$$G_{ij}^{(4)} = \frac{1}{4}G_{ik}^{(2)}G^{(0)kl}G_{lj}^{(2)}. \quad (4)$$

In the standard AdS/CFT dictionary, Dirichlet boundary conditions are imposed by holding $G^{(0)}$ fixed on a UV cutoff surface such as $\rho = \epsilon$. The subleading metric falloff $G^{(2)}$ maps to the stress tensor one-point function in the dual field theory and, up to conservation and tracelessness, is unconstrained. Now consider a particular fixed asymptotically AdS$_3$ spacetime, with metric denoted by $G$. This fixed metric $G$ is the solution to Einstein's equations for a set of not one but many different boundary conditions at the asymptotic boundary. A special set of such boundary conditions parametrised by a variable $\lambda$ holds fixed at $\rho = \epsilon$ the combination

$$G_{ij}^{(0)} - \frac{\lambda}{4\pi G_N L}G_{ij}^{(2)} + \left(\frac{\lambda}{4\pi G_N L}\right)^2 G_{ij}^{(4)}. \quad (5)$$

The boundary conditions (5) are mixed; they hold a combination of the metric and its derivatives fixed. As an example of how a given metric can be the solution to many different boundary conditions take $G$ equal to be pure AdS$_3$, for which $G^{(2)} = G^{(4)} = 0$ and $G_{ij}^{(0)} = \eta_{ij}$. This metric $G$ satisfies the mixed boundary conditions (5) for all $\lambda$, if the combination is fixed to $\eta_{ij}$.

The mixed boundary conditions (5) for the bulk metric at the asymptotic boundary are special [39]. In the bulk this is because they are equivalent to Dirichlet boundary conditions at finite radial cutoff, i.e. holding fixed the induced metric $G_{ij}(\rho_c, x)$ on a finite radial cutoff surface

$$\rho_c = -\frac{\lambda}{4\pi G_N L}. \quad (6)$$

This can be easily checked from the Fefferman-Graham expansion (3). Dirichlet boundary conditions at finite cutoff are interesting because then the physics in the interior of the cutoff surface is an effective description of semiclassical gravity in a spatial volume that is finite, unlike AdS [38]. The bulk gravitational action evaluated in the whole bulk does not depend on $\lambda$, but the set of on-shell metric configurations does. Since the mixed metric boundary conditions at the asymptotic boundary are equivalent to Dirichlet boundary conditions with metric equal to that which $G$ induces on a finite radial cutoff surface, the gravitational theories with these different boundary conditions all by construction include $G$ amongst their on-shell metric configurations, but in general have little other overlap in on-shell metric configurations.

In the dual holographic field theory, it is an old story that replacing Dirichlet with mixed boundary conditions for the bulk metric at infinity maps to deforming the original undeformed

---

[2]The coordinate $\rho$ is related to the Poincaré coordinate $z$ by $\rho = z^2$.

field theory with a double trace deformation [45]. The particular double-trace deformation that the mixed boundary conditions (5) corresponds to is the $T\overline{T}$ deformation. Specifying a bulk metric $G$ and mixed boundary condition parameter $\lambda$ is sufficient to fix the dual $T\overline{T}$-deformed holographic CFT living on a manifold $M$. The field theory background metric on $M$ is most simply expressed in terms of the metric induced by $G$ on the $\rho = \rho_c$ surface denoted $\tilde{M}$:

$$\gamma_{ij}(x) = \rho_c\, G_{ij}(\rho_c, x) = \rho_c G|_{\tilde{M}}\,. \tag{7}$$

The metric on $M$ is $\gamma$, and the metric on $\tilde{M}$ is $g$. Note that, from this map between metrics, a subregion of $M$ is empty iff the corresponding subregion of $\tilde{M}$ is also empty. In general $\gamma$ has a non-trivial $\lambda$ dependence. The action of the deformed theory is the solution to the flow equation

$$\frac{d}{d\lambda}S^{(\lambda)} = \int d^2x\, \sqrt{\gamma}\, \mathcal{O}^{(\lambda)}_{T\overline{T}} \quad \text{with} \quad \mathcal{O}^{(\lambda)}_{T\overline{T}} := -\frac{1}{2}(\gamma_{ik}\gamma_{jl} - \gamma_{ij}\gamma_{kl})T^{ij}T^{kl}\,. \tag{8}$$

The seed action is that of the undeformed CFT, $S^{(0)} = S_{CFT}$, which is dual to the gravitational theory with asymptotic Dirichlet boundary conditions. The deforming operator $\mathcal{O}^{(\lambda)}_{T\overline{T}}$ has a $\lambda$ dependence because the stress tensor changes as the action flows. From the bulk gravitational perspective, what is special about the $T\overline{T}$ deformation in the dual holographic CFT is that the corresponding mixed boundary conditions are equivalent to Dirichlet boundary conditions on a finite radial surface.

### 2.1.2 Coarse-grained description

Here we introduce the coarse-grained version of holographic $T\overline{T}$: the effective description of gravity within a finite box, and the EFT of a $T\overline{T}$ deformed CFT. Consider a gravitational partition function with Dirichlet boundary conditions $Z_{grav}[g]$, which depends on the value of the induced metric $g$ on its boundary. As a consequence of diffeomorphism invariance, this dependence is constrained by (a radial version of) the Wheeler-DeWitt (WDW) equation [46]

$$H_{WDW}Z_{grav}[g] = 0\,, \tag{9}$$

where $H_{WDW}$ is the WDW Hamiltonian[3]

$$H_{WDW} = g_{ij}\frac{\delta}{\delta g_{ij}} + \frac{1}{\sqrt{g}}(g_{ik}g_{jl} - \frac{1}{2}g_{ij}g_{kl})\frac{\delta}{\delta g_{ij}}\frac{\delta}{\delta g_{kl}} + \sqrt{g}R\,. \tag{10}$$

Formally this equation can be solved to relate gravitational partition functions with different metric boundary conditions:

$$Z_{grav}[g] = \int D\tilde{g}\, \mathcal{K}[g, \tilde{g}]Z_{grav}[\tilde{g}]\,. \tag{11}$$

The kernel can be calculated for any bulk field content, including with matter [48, 49], but we won't need the precise form for our discussion. The initial data we want to use when solving (11) is the gravitational partition function with AdS Dirichlet boundary conditions, i.e. the undeformed CFT generating functional. Suppose we take an asymptotically AdS metric $G$ and set $g$ equal to the metric induced on a radial cutoff surface $\tilde{M}(\lambda)|_G$ parametrised by $\lambda$. With

---

[3]The quantised WDW Hamiltonian has contact terms that can be removed by normal ordering but which leave operator ordering ambiguities in its definition [47]. These are similar to the operator ordering ambiguities arising from the point-splitting regularisation of the $T\overline{T}$ operator. Since the primary focus of this paper is on cost proposals rather than $T\overline{T}$ we will not address these subtleties here.

(11) we have an effective description $Z_{grav}[g = G|_{\tilde{M}(\lambda)}]$ of gravity with Dirichlet boundary conditions that fix the induced metric equal to the metric induced on $\tilde{M}(\lambda)$ by $G$. The effective description 'throws away' everything outside the cutoff surface. $Z_{grav}[g = G|_{\tilde{M}(\lambda)}]$ has the geometry of $\text{Int}(\tilde{M}) \cap G$ as one allowed on-shell metric configuration among many. Other on-shell metric configurations differ at the boundary in their normal derivatives, corresponding to different one-point functions in the dual cut-off theory.

Gravity with a Dirichlet metric boundary condition $g = G|_{\tilde{M}(\lambda)}$ has a dual non-gravitational description as the effective theory of a $T\overline{T}$-deformed CFT on a manifold $M$ with degrees of freedom above the scale set by $\lambda$ integrated out [38, 42]. The map between EFT generating functional and gravitational partition function is

$$Z_{EFT}^{(\lambda)}(\gamma) = Z_{grav}[g = G|_{\tilde{M}(\lambda)}]. \tag{12}$$

This is a continuous family of EFT's parameterised by $\lambda$ with non-dynamical background metrics $\gamma$.

What is the map from the bulk metric boundary condition $g$ to quantities in the dual non-gravitational EFT? The metric $g$ fixes the conformal class of the field theory background metric:

$$\gamma_{ij}(x) = \lambda^{2/d}(x)g_{ij}(x), \tag{13}$$

with the deformation parameter related to the radial coordinate in Fefferman-Graham gauge by $\lambda(x) \propto -\rho_c(x)^{d/2}$ [48]. In general we can have a non-constant deformation, corresponding to a non-constant radial cutoff. The holographic $T\overline{T}$ deformation in dimensions other than two and with sources turned on is defined by the property that it is dual to a bulk gravitational theory with Dirichlet conditions at finite cutoff. The $T\overline{T}$ deformed non-gravitational effective action is formally the solution to the flow equation

$$\frac{\delta}{\delta\lambda(x,t)} \log Z_{EFT}^{(\lambda)}(\gamma) = \int_M d^d x \, \langle X^{(\lambda,\gamma)} \rangle . \tag{14}$$

This flow equation, including the precise form for $X$, is derived by mapping the gravitational WDW equation (9) to an exact RG equation in the field theory [38, 42, 48]. Schematically, first we take the semiclassical limit where the gravitational partitional function is given by the on-shell gravitational action. Then, functional variations $\delta_g Z_{grav}[g]$ with respect to $g$ become functions of the Brown-York stress tensor of the on-shell action, and in the boundary effective action this maps to deformations by the field theory stress tensor. In the special case of 2d, with a constant deformation parameter $\lambda$, and with no sources except a flat background metric turned on, the deformation is the standard $T\overline{T}$ operator, $X = \mathcal{O}_{T\overline{T}}$ defined in (8).

## 2.2 Bulk path integrals

We have a one-to-many map from a Cauchy slice of fixed bulk spacetime $G$ to a geometric state in the Hilbert spaces of not one but a continuous family of $T\overline{T}$-deformed holographic CFTs. Each mixed boundary condition for which $G$ is an on-shell solution maps to a different $T\overline{T}$ deformed theory. The whole of $G$ maps to the causal development of these geometric states. We thus have many field theoretic descriptions for the causal evolution of a spatial slice of $G$. This leads to the key question that largely motivates our paper: which of the deformed CFTs describes the evolution of the bulk state in the least 'complex' way?

To be concrete, let us focus on transition amplitudes between geometric states with metric and conjugate momentum induced by a given fixed $G$ on two arbitrary Cauchy slices $\Pi_1$ and $\Pi_2$, see figure 1. By construction $G$ is the dominant spacetime saddlepoint contribution in the

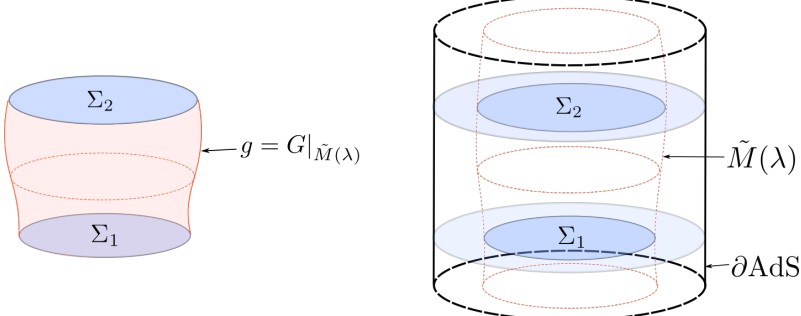

Figure 1: Coarse-grained vs fine-grained bulk path integrals. Left: A path integral representation of the transition amplitude between two compact manifolds $\Sigma_1$ and $\Sigma_2$ with specified metrics. The gravitational path integral has Dirichlet boundary conditions $g$. Right: Embedding in a larger UV complete theory of gravity in AdS. Fix a bulk metric $G$, which is a solution to a family of mixed boundary conditions at the asymptotic boundary parameterised by $\lambda$. The mixed boundary conditions are on-shell equivalent to Dirichlet boundary conditons on surface $\tilde{M}(\lambda)$. The light blue surfaces are Cauchy slices $\Pi_1$ and $\Pi_2$, which $\Sigma_1$ and $\Sigma_2$ are compact subregions of.

semiclassical approximation to the transition amplitude between those two geometric states. This transition amplitude has a formal path integral representation

$$\langle \psi_2 | \psi_1 \rangle = \int D\tilde{G} \, e^{iS_{grav}[\tilde{G}]} \delta \left( \tilde{G}|_{\Pi_1} - G|_{\Pi_1} \right) \delta \left( \tilde{G}|_{\Pi_2} - G|_{\Pi_2} \right). \tag{15}$$

The $\delta$-functions are the wavefunctions of the initial and final states in the basis of geometric states, and they in effect impose spacelike boundary conditions on the path integral. We also implicitly impose the timelike boundary condition (5). Finding the classical gravitational solution for these boundary conditions is not an inconsistent overdetermined problem because $G$ is by construction a solution. $S_{grav}$ is the Einstein-Hilbert action including, if necessary, boundary, corner, and holographic counterterms [50, 51].

We would like to interpret the bulk path integral (15) and its dominant saddlepoint $G$ as originating from the continuum limit of a tensor network representation of the transition amplitude, and associate a cost, i.e. a properly understood regularised number of gates, to that tensor network. Keeping $G$ fixed and changing the asymptotic boundary conditions does not affect the gravitational transition amplitude (15) in the saddle-point limit,[4] but are the costs of the associated path integrals the same? We propose that the 'fundamental gates' the gravitational theories use are in fact different, and so the cost of their path integrals – even those that prepare the same final state from a given initial state – may be different. This is backed up by the fact that each mixed boundary condition maps to a field theory with different $T\overline{T}$-deformation parameter, which at least naively have different operators and gates available to them [39, 52, 53]. This introduces a $\lambda$ dependence to the 'cost' of $G$, which we can minimise over in our pursuit of finding the least complex way of evolving between two geometries.

Both $T\overline{T}$-deformed CFTs and gravitational duals are thought to be UV complete [38, 39, 52, 54], so we expect UV divergences in state complexity and the cost of preparing or evolving between states. This is associated to the infinite volume of all spatial slices of all asymptotically AdS spacetimes. To regulate we need to coarse-grain and consider effective theories with a cutoff. In principle we could choose any UV cutoff as our regulator, but $T\overline{T}$-deformed theories and their generalizations come with a natural effective UV scale, which is the scale above

---

[4]There are subleading in $1/N$ corrections to this statement from the metric and matter field fluctuations which we neglect. Our proposals for path integral cost are only accurate to leading order in $1/N$.

which the theory becomes approximately non-local. It is therefore very natural to integrate out all degrees of freedom above this scale, but keep the degrees of freedom below this scale, as the same scale is also expected to set the effective size of the smallest possible tensor in a tensor network description. In the $T\overline{T}$ example discussed above the relevant scale is set by the deformation parameter $\lambda$. In the bulk, there is a natural dual description of this UV cutoff, namely the existence of the Dirichlet surface $\tilde{M}(\lambda)$. In effect $\tilde{M}(\lambda)$ splits the bulk spacetime $G$ into an exterior UV region, and an interior IR region, and we 'integrate out' the exterior. This is an important point so let us emphasize this once more: *The bulk Dirichlet surface defines both the boundary conditions for the dual field theory, as well as the UV cutoff in that theory.*

We will make this more precise: The path integral depicted by figure 1 is the transition amplitude between two spatial geometries on $\Sigma_1$ and $\Sigma_2$ in an effective gravitational theory $Z_{grav}[g]$ which holds the boundary metric fixed to $g$. This can be embedded in a UV complete theory of gravity in AdS if there exists a $G$ and a set of asymptotic boundary conditions parametrised by $\lambda$ such that $g = G|_{\tilde{M}(\lambda)}$, and $\Sigma_1$ and $\Sigma_2$ can be embedded in Cauchy slices of $G$, i.e. the metrics on $\Sigma_1$ and $\Sigma_2$ equal the metrics $G$ induces on $\Sigma_1 \subset \Pi_1$ and $\Sigma_2 \subset \Pi_2$. Then by construction the dominant saddlepoint geometry in the effective theory coincides with $G$ restricted to the interior of $\tilde{M}(\lambda)$. To 'optimise' the path integral we keep $G$ and the compact submanifolds $\Sigma_1$ and $\Sigma_2$ fixed, and vary $\lambda$. Pictorially, this amounts to varying the shape of the red surface in the left half of figure 1 while keeping on-shell interior geometry equal to the geometry inside $\tilde{M}(\lambda)$ in the right half of the figure. This gives us a continuous family of gravitational path integrals that prepare the same final bulk state $|\Sigma_2\rangle$ from a given reference state $|\Sigma_1\rangle$.[5] Note that to keep $\Sigma_1$ and $\Sigma_2$ fixed and $\lambda$-independent we need the intersection of $\tilde{M}(\lambda)$ with $\Pi_1$ and $\Pi_2$ to be $\lambda$-independent, which in general requires $\tilde{M}(\lambda)$ to have non-constant radius, which in turn requires a non-constant $\lambda$ parameter [48].

We have given the explicit map between the bulk metric $g$ on $\tilde{M}$ and the background field theory metric $\gamma$ on $M$ in (13). If we wish we can use the same coordinates $x^i$ on both manifolds. By extension, given a bulk path integral between Cauchy slices as depicted in figure 1 we know how to map between intersections of those bulk Cauchy slices with $\tilde{M}$ and boundary Cauchy slices of $M$. From this we have a complete and precise holographic duality between a given path integral in an effective gravitational theory and a path integral in a $T\overline{T}$-deformed CFT. In a slight abuse of notation, when discussing path integrals on the subregions of $M$ and $\tilde{M}$ between Cauchy slices, we will also call these subregions $M$ and $\tilde{M}$. While in the remaining part of the paper we will not make an explicit use of the above discussion about holographic $T\overline{T}$, we believe it can be taken as a starting point in building a bridge between gravitational notions of cost functions and operatorial expressions for circuits living on $\tilde{M}(\lambda)$.

We consider bulk path integrals in both Lorentzian or Euclidean signature. In some ways they are similar: for both signatures the path integrals on manifolds with one boundary prepare states, and when there are two boundaries the path integrals calculate transition amplitudes. Lorentzian and Euclidean bulks differ however in the possible metric signatures of their embedded surfaces. For Euclidean bulks all surfaces $\tilde{M}$ are spacelike. Path integrals on $\tilde{M}$ with two boundaries correspond to unnormalised density matrices,

$$\rho = P e^{-\int d\tau H(\tau)}, \tag{16}$$

with infinite Euclidean time evolution preparing the projection operator onto the vacuum state $\rho = |0\rangle\langle 0|$. For Lorentzian bulks the embedded surface $\tilde{M}$ can be spacelike, timelike, or even non-constant signature. Since the signature of $\tilde{M}$ is the same as $M$, as follows from (13), in the $T\overline{T}$ deformed boundary theory a timelike $\tilde{M}$ corresponds to a Lorentzian path integral while spacelike gives Euclidean ones. States on time-reflection symmetric slices in Lorentzian AdS

---

[5]Herein, we assume both of these states to live effectively in the IR-sector of the Hilbert space of the UV-complete theory that is created by coarse graining.

spacetimes can be prepared by a Euclidean path integral, while other states cannot and need some Lorentzian time evolution.

We consider path integral cost proposals for both Euclidean and Lorentzian bulks. A cost proposal that is physically reasonable, e.g. non-negative, in one signature need not be in the other, so part of specifying a cost proposal is saying whether it is applicable to Lorentzian or Euclidean bulks, or both. When we come to giving specific cost proposals we will always specify which metric signature it is applicable to. The question as far as path integral optimisation of state preparation goes is which set of $\tilde{M}$ one is minimising cost over. For Lorentzian bulks if one wishes to find the shortest path in a space of *unitary* operators, then one should restrict to only timelike $\tilde{M}$. Spacelike components of $\tilde{M}$ in Lorentzian bulks that are achronal with respect to each other and $\Sigma_1$ and $\Sigma_2$ can be thought of as extensions of the partial Cauchy slices on which the initial and final states are defined. Whether they are part of the past or future slice depends on the time-orientation of their normals. The states on $\Sigma_1$ and $\Sigma_2$ are then reduced state with respect to the larger slices. In this paper we will only consider timelike $\tilde{M}$ in Lorentzian bulks.

One last closing comment is in order. In the present work we start with an asymptotically AdS geometry and identify in it the cut-off surface $\tilde{M}(\lambda)$ and its interior, as in figure 1 (right). Starting with the situation depicted in figure 1 (left), it is a priori not clear if it can be embedded in an asymptotically (perhaps locally) AdS geometry. There are several reasons for it. One is a possible issue of singularities arising as one tries to extend the geometry towards infinity and another are matter fields that may enforce non-AdS asymptotics. A special case is pure gravity with negative cosmological constant in three bulk dimensions, in which case the geometry is guaranteed to be a portion of the $\text{AdS}_3$ manifold.

## 2.3 Path integral optimisation and holographic state complexity

Suppose we have a prescription for associating a cost to the gravitational path integral depicted in the left half of figure 1. Heuristically, this cost denoted $\mathscr{C}(\tilde{M}(\lambda))$ 'counts' the number of gates in different spacetime tensor networks parametrised by $\lambda(x, t)$. Each $\lambda(x, t)$ defines a different path integral that maps the same fixed initial state $|\Sigma_1\rangle$ to final state $|\Sigma_2\rangle$, and a given one of these bulk path integrals will not correspond to the least complex circuit between those states, which is why we are discussing cost rather than complexity.

There is however a sense in which the bulk path integrals can be optimised, by minimising path integral cost $\mathscr{C}(\tilde{M}(\lambda))$ over $\lambda$. Recall that we keep $G$, $\Sigma_1$ and $\Sigma_2$ fixed, so the set of $\lambda$ to minimise over are those for which

$$\partial \tilde{M}(\lambda) = \partial \Sigma_1 \cup \partial \Sigma_2. \tag{17}$$

In some cases the minimal path integral cost can be interpreted as state complexity, and this allows us to connect holographic path integral cost to holographic state complexity. The subtlety is in which set of path integrals it is meaningful to compare.

Stepping back for a moment, when does it make sense to optimise path integrals? Path integrals in a fixed seed field theory but allowing for different background geometries, field theory sources and field theory deformations define different operators, and it is not meaningful to compare the costs of the path integrals. Minimising cost over all such path integrals cannot meaningfully be interpreted as 'optimisation' of state preparation, if indeed they even act on the same Hilbert spaces. The key idea is that there are a set of bulk path integrals it is meaningful to optimise: those that take a given initial state to the same final state.

To connect path integral cost to state complexity, we also need to choose a suitable initial state, such that the final state is prepared from 'nothing'. In the complexity literature this is often taken to be a spatially unentangled product state, but this does not have an approximate

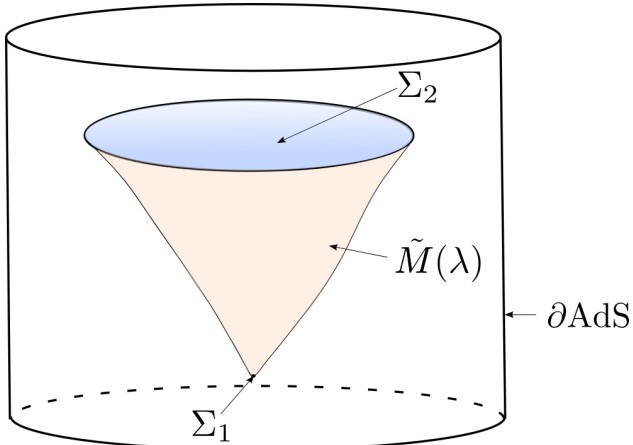

Figure 2: A fine-grained description of the preparation of a bulk state from 'nothing'. The initial state lives on a surface $\Sigma_1$ embedded deep in the IR of an asymptotically AdS spacetime. This corresponds to a state in a $T\overline{T}$-deformed CFT with very large deformation parameter. We take this to be the reference state when defining the state complexity of the final state on $\Sigma_2$.

semiclassical description, so we cannot use it. Our initial state takes $\Sigma_1$ to be a bulk point. It is helpful to think of this as limit of a small spatial region placed deep in the IR of the asymptotically AdS spacetime, which we shrink to zero volume, see figure 2. As $\tilde{M}(\lambda)$ moves outwards from the deep IR towards $\partial\Sigma_2$, the Hilbert space of the effective gravitational theory grows to non-trivial size. In the dual boundary theory this initial state lives in a theory where we have taken the $T\overline{T}$ deformation parameter $\lambda \to \infty$. This sets the effective RG scale to zero, and the effective theory integrates out everything above that scale which leaves a trivial theory.

The bridge from path integral cost to state complexity is still not complete. Even the optimum path integral may not correspond to the shortest path in the space of states, because not all unitary operators are generated by the set of path integrals we have considered. This means that we only expect the cost of the optimum path integral to still only upper bound the complexity of state $|\Sigma_2\rangle$:

$$\min_{\lambda} \mathscr{C}(\tilde{M}(\lambda)) \geq C(|\Sigma_2\rangle). \tag{18}$$

It is not clear when if ever this inequality is saturated, i.e. when if ever the least complex unitary operator taking $|\Sigma_1\rangle$ to $|\Sigma_2\rangle$ has a path integral representation.

The semiclassical bulk path integrals and their associated costs are defined on a specified and fixed bulk geometry, but the set of path integrals we want to optimise over do not necessarily have the same background geometry. We can choose to keep the bulk fixed and optimise over the set of path integrals on subregions of that single fixed bulk defined by their boundary $\Sigma_1 \cup \tilde{M}(\lambda) \cup \Sigma_2$. A larger and more natural superset allows for different bulk geometries $G$, as well as different $\tilde{M}(\lambda)$, that also keep the initial and final states $|\Sigma_1\rangle$ and $|\Sigma_2\rangle$ fixed. When we come to connecting to holographic state complexity proposals we will choose, for the sake of simplicity, path integral cost proposals for which we can show that the minimal cost is the same for either set.

This concludes our discussion of transition amplitudes between bulk states. We have discussed what Hilbert spaces the bulk states live in, what the exact map is from bulk theory and state to $T\overline{T}$-deformed CFT and state, constructed a set of path integrals we can 'optimise' over that prepare the same state from an initial state, how to coarse-grain the description on both sides, and how to prepare a state from nothing. Next we will consider proposals for the cost

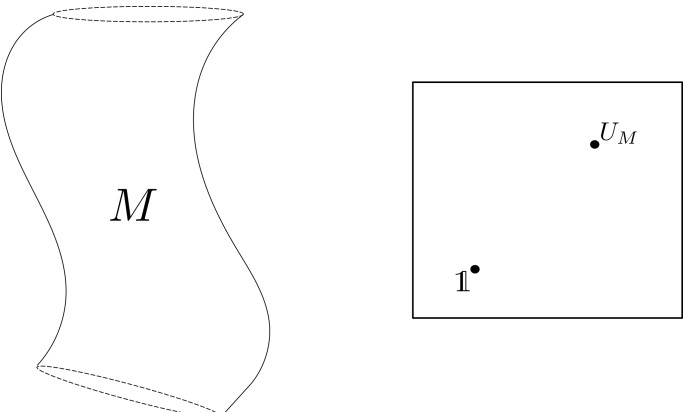

Figure 3: Left: $M$ is a Euclidean or Lorentzian manifold with two boundaries from which operators can be defined. Right: The space of all such operators, which are unitary when $M$ is Lorentzian. $U_M$ is the operator whose matrix elements are calculated by the path integral on $M$.

$\mathscr{C}(\tilde{M}(\lambda))$ of the coarse-grained bulk path integrals, which gives us a quantity to minimise and so optimise.

## 3 Holographic path integral cost proposals

In AdS/CFT the boundary path integral defines a (Lorentzian/Euclidean) time-evolution operator. The goal of this section is to consider holographic proposals for the cost of the path integral, i.e. the length of the path through the space of operators.

### 3.1 Path integral cost

Let us start by defining what we mean by path integral cost. This subsection is mostly field theoretic and logically separate from gravity and holography. We may associate an operator $U_M$ to the path integral on any manifold $M$ with two boundaries and unspecified boundary conditions, see figure 3. Matrix elements of the operator are defined by the path integral with specified boundary conditions for the fields of interest $\phi = \phi_{1,2}$ on the two boundaries, which computes a transition amplitude:

$$\langle \phi_2 | U_M | \phi_1 \rangle := \int_{\phi=\phi_1}^{\phi=\phi_2} D\phi \, e^{-S[\phi]}. \tag{19}$$

We are interested in both Euclidean and Lorentzian path integrals. For the latter there is an insertion of $i$ in the path integral representation of the transition amplitude. We denote the operator defined in (19) by $U_M$ whether or not the operator is unitary, i.e. even if the path integral is Euclidean. $U_M$ depends not only on the geometry of $M$, but on the field theory itself. We take the field theory to be holographic and consider not only changes to the background geometry but also allow the addition of sources and deformations to the theory. When we come to embedding $M$ in a bulk spacetime, these sources and deformations will correspond to adding bulk excitations and bringing in the boundary to finite cutoff with $T\overline{T}$ deformations. From the context we hope it is clear whether $M$ refers only to the manifold, or to all the data required to define the path integral including the manifold, seed theory, sources, and theory deformations.

We define path integral cost $\mathscr{C}$ as the length of the path generated by time evolution from the identity operator to $U_M$ in a metric space of operators. For the path integral cost to be well-defined the metric space must be specified: both which set of operators to include and which metric to impose on it. Expressed in the terminology of Nielsen's geometric formulation [18], cost functions determine the metric on the set of operators, and control functions specify paths in that metric space. The path integral generates a path from $\mathbb{1}$ to $U_M$ which is not generally geodesic, and so the path integral cost upper bounds the operator complexity of $U_M$, because that is defined as the length of the *shortest* path between $\mathbb{1}$ and $U_M$. An example illustrating the difference between path integral cost and operator complexity is given in figure 4. The operator complexity of $U_M$ itself upper bounds the state complexity of the state $|f\rangle = U_M |i\rangle$, because state complexity can be defined as operator complexity minimised over those unitary operators that map a fixed reference state $|i\rangle$ to the target state $|f\rangle$. The path integral operator $U_M$ does not in general correspond to the shortest path between $|i\rangle$ and $|f\rangle$, which is why its complexity is only an upper bound to the state complexity. These statements can be summarised as

$$\mathscr{C}(M) \geq C(U_M) \geq C(|f\rangle), \tag{20}$$

where $\mathscr{C}(M)$ is the path integral cost of $M$, $C(U_M)$ the operator complexity of $U_M$ and $C(|f\rangle)$ the state complexity of $|f\rangle = U_M |i\rangle$. Note the calligraphic font that distinguishes cost $\mathscr{C}$ from complexity $C$.

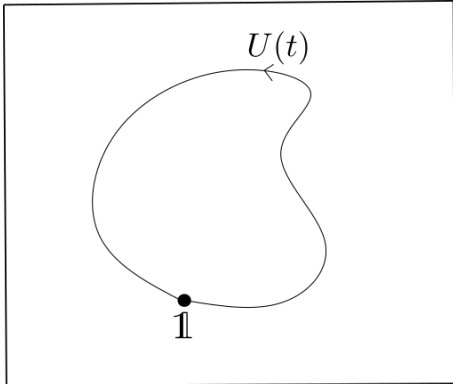

Figure 4: An example illustrating the difference between path integral cost and operator complexity. Suppose we have a manifold $M$ such that Hamiltonian evolution from the initial to final boundary traces out a closed path through the space of unitary operators, i.e. a Poincaré recurrence with $e^{iHt_f} \approx \mathbb{1}$. The path integral cost is the length of the closed path, which is non-zero, while the complexity of the time evolution operator is trivially zero. This is an example where $\mathscr{C}(M) \neq C(U_M)$.

Note that the same operator $U_M$ can be represented in an infinite number of ways as a circuit in physical time upon picking a time foliation, see figure 5. From the circuit perspective, constant time slices can be thought of as layers of the circuit, and these different time foliations as different ways of assigning gates amongst the layers. The circuit as a whole is independent of its time foliation, and this is a physical reason for why the cost should be foliation independent. This we impose on the bulk side of the holographic proposal through a covariance requirement. On the boundary side one implication is that the lengths of the red, blue and other paths in figure 5 from $\mathbb{1}$ to $U_M$ are the same. Hence, while in general two randomly selected paths connecting $\mathbb{1}$ and $U_M$ will have different lengths (due to describing physically different circuits), each such path will come with an equivalence class of paths of the same length that can be generated by changes in time-foliation. This is a symmetry of the metric space that is a consequence of the physical equivalence of different time foliations.

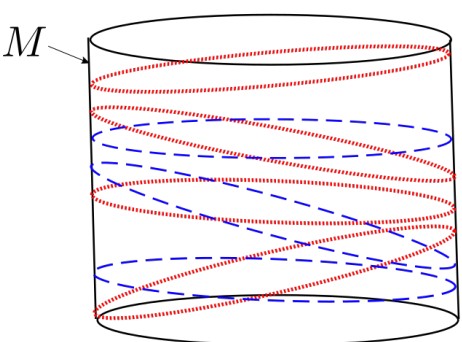
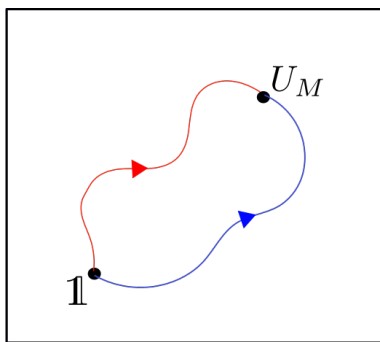

Figure 5: Left: $M$ is taken to be a Lorentzian cylinder, and the blue and red ellipses represent constant time slices of two different time foliations of the cylinder. Right: $U_M$ is the unitary operator whose matrix elements are calculated by the path integral on $M$. Different time foliations define different paths to $U_M$.

## 3.2 Physical properties of path integral cost

Necessary conditions that holographic proposals must satisfy in order to be reasonably interpreted as path integral cost, i.e. the length of a path in a metric space of operators, are the following:

1. The trivial path integral has zero cost.

   When $M = \varnothing$ the path integral is trivial with no time evolution, the operator associated to it is the identity, and holographic proposals should evaluate to zero cost: $\mathscr{C} = 0$. We can, and will, even strengthen this requirement by demanding that the trivial path integral is the *only* one that has zero cost.

2. Additivity.

   Concatenating path integrals joins paths in the space of operators, and cost is the total length of the path, so cost is additive. This means that if we have $M$ and $M'$ which share a boundary, then $\mathscr{C}(M \cup M') = \mathscr{C}(M) + \mathscr{C}(M')$. This distinguishes cost from complexity which is subadditive: $C(U_M \cdot U_{M'}) \leq C(U_M) + C(U_{M'})$.

3. Symmetry.

   The length of a path traced through a metric space of operators from $A$ to $B$ is the same as from $B$ to $A$. This means that holographic proposals for path integral cost cannot depend on which way around the two connected components of $\partial M$ are labeled the 'initial' and 'final' boundaries.

4. Covariance.

   The cost of a path integral on a manifold should be independent of the coordinates used to describe the manifold.

5. Non-negativity.

   A discretised path integral is a circuit, and path integral cost is a measure of the number of gates in that circuit. This number cannot be negative so path integral cost must be non-negative.

The above are essential points in order to sensibly associate path integral cost to spacetime regions in holography. In relation with the properties of the existing holographic complexity

proposals, one may also want to impose that for TFD states and their gravitational representation in terms of eternal black holes, optimal paths in our proposals give rise to late-time linear growth and switchback effect [9]. These effects are only conditions in Lorentzian setups and on TFD states, not on every state like the other requirements, so for us they are not as fundamental as items 1-5 above.

## 3.3 The space of all proposals: from boundary path integrals to functions on bulk subregions

We are looking for proposals for the gravitational dual of the cost of a holographic field theory path integral. Naturally it is equally valid to think of the path integral in the gravitational or non-gravitational description, and we have given the explicit map between the two, but quantities such as cost need not manifest the same way on the two sides. The set of cost proposals we consider take inspiration from existing holographic state complexity proposals and have two aspects in common: (1) a geometric map from the subregion of $M$ or $\tilde{M}$ on which the path integral is defined to bulk subregion $X_M$, and (2) a function $f(X_M)$ on that bulk subregion. These two shared aspects take inspiration from existing holographic state complexity proposals, which are also functions on bulk subregions.

We want to consider all such pairs of maps which together define a tentative holographic cost proposal:

$$\mathscr{C}(M) = f(X_M). \tag{21}$$

The set of cost proposals we start from contains an infinite number of ways of specifying $X_M$ given $M$, and an infinite number of functions $f$. Cost, the length of path in a metric space, obeys certain mathematical and physical properties, and we will see the extent to which the space of possible gravitational duals can be reduced by imposing these properties.

### 3.3.1 Specifying the bulk subregion: $M \to X_M$

We want to work within a single fixed bulk spacetime. As discussed in section 2, fixing which mixed boundary conditions to use in the bulk theory fixes the deformation parameter $\lambda$ in the boundary theory, and fixing the bulk geometry $G$ fixes the actual deformation, the background field theory sources including the metric, and the boundary state including its causal evolution. Functions on bulk subregions of a fixed $G$ can only be dual to the cost of the boundary path integral that corresponds to Hamiltonian evolution of the boundary state dual to that bulk geometric state, rather than an arbitrary path integral in the boundary theory.

The two boundaries of $\tilde{M}$ have to be attached to the hypersurfaces $\Sigma_1$ and $\Sigma_2$ as in figure 1, which should be parts of bulk Cauchy slices and hence achronal. In section 2 these hypersurfaces were specified and fixed, but in this section where we start from the boundary theory with a specified fixed $M$ they are not unambiguously defined. For the bulk path integral the choice of $\Sigma_1$ and $\Sigma_2$ is as fundamental as the choice of $\tilde{M}$, but we may still wonder whether in a given prescription these can be defined according to some unambiguous and covariant rule. Some examples of ways to define hypersurfaces $\Sigma_1$ and $\Sigma_2$ that we attach to the boundaries of $\tilde{M}$ are, in increasing degree of generality,

1. Future/past directed null surfaces, as in the CA and CV2.0 proposals.

2. The extrema of some functional defined on the hypersurface, as in the CV proposal.

3. The solution to $\xi = 0$, where $\xi$ is some function of the local intrinsic and extrinsic geometry. This need not be the extremum of any given functional.

4. The same as 3., except now allowing for non-local and global data in the definition. An example would be to define the hypersurfaces as the solution to $K_\Sigma = \text{Vol}(\tilde{M})$.

This list includes the codimension-one surfaces appearing in holographic complexity proposals, and a large family of generalisations, though the list is not exhaustive.

Consider the bulk spacetime subregions $X_M$ that can be covariantly defined with respect to a codimension-1 surface $\tilde{M}$. There are of course an infinite number of such prescriptions; let us first consider those that are similar in nature to the holographic complexity proposals. One natural candidate is the interior of $\Sigma_1 \cup \tilde{M} \cup \Sigma_2$ which is codimension-0 and which we label $N$, see figure 6. Codimension-0 bulk regions are used in the CA and CV2.0 state complexity proposals. We can also take $X_M$ to be codimension-1 with respect to the bulk. Natural candidates include $\tilde{M}$, $\Sigma_1$ or $\Sigma_2$, or any union of these. We will show later how taking $X_M = \tilde{M}$ and evaluating its volume gives a cost proposal that when 'optimised' reduces to the CV state complexity proposal.

These candidates for $X_M$ only scratch the surface of possibilities. At this stage there is nothing to favour one candidate over another; it is only when we impose the physical properties of cost that we can rule out possibilities.

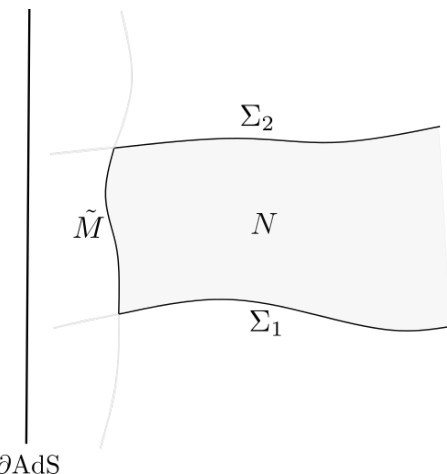

Figure 6: We are looking for holographic proposals for the cost of the path integral on $M$. $\tilde{M}$, $\Sigma_1$, $\Sigma_2$ and $N$ are representative of bulk subregions of various codimension that can be covariantly defined and on which functions can be evaluated as part of a holographic proposal.

### 3.3.2   Functions on bulk subregions

With a specified bulk region $X_M$, we may propose the cost $\mathscr{C}(M)$ of the path integral on $M$ to be a function evaluated on that region: $f(X_M)$. To complete the cost proposal the function $f$ needs to be specified. Some examples in order of decreasing simplicity include the region's:

1. Volume,

2. Gravitational action, including the Einstein-Hilbert term on codimension-zero regions, the Gibbons-Hawking-York (GHY) boundary term on non-null codimension-one regions, Hayward-type corner terms on codimension-two regions, or appropriate terms on null-boundaries, see e.g. [55],

3. Local functionals of curvature invariants: $f = \int \sqrt{|g|}\xi(R_{\mu\nu\rho\sigma}, K_{mn})$,

4. More general local functionals: $f = \int \sqrt{|g|}\xi(R, R^2, R_{\mu\nu}R^{\mu\nu}, ..., K, K^2, ..., \phi, ...)$, where, for example, $\phi(x)$ could be some non-physical auxiliary scalar field that appears in the complexity proposal but not in the bulk Lagrangian,

5. Non-local functionals, e.g. $f = \int d^d x d^d y R(x) R(y)$.

Again, these candidates for functions on bulk subregions only scratch the surface of possibilities, and except for appeals to simplicity there is nothing to favour one candidate over another until we impose the physical requirements of computational cost.

## 3.4 Reducing the space of cost proposals

We are considering the set of holographic proposals for the cost of a boundary path integral, which consists of a map from the surface on which the path integral is defined to a bulk subregion, and a function on that subregion. Let us apply the physical requirements of section 3.2 to identify which functions on bulk subregions can be interpreted as cost.

1. Only the trivial path integral has zero cost.

   This requires $f(X_M) = 0$ when $M = \tilde{M} = \varnothing$. This rules out for example $\mathrm{Vol}(\tilde{M})$ for Lorentzian bulks, because we can have an $\tilde{M}$ that is non-empty but has zero volume because it is a null surface. The same cost proposal applied to Euclidean bulks *is* allowed.

   This requirement fixes additive constants. It does not rule out any proposal of the form (22) if $X_M = N$ as long as the integrand is non-singular and $N \to 0$ in that limit. This rules out some $X_M$, such as $\Sigma_1$ being past-directed null and $\Sigma_2$ future-directed null.

2. Additivity.

   Additivity does not rule out any proposal that is the integral of a local density, such as volume or gravitational action, as long as the contribution from $X_{M_1} \cap X_{M_2}$ vanishes. A non-trivial example of how this can occur is when $f$ includes GHY boundary terms, because if $X_{M_1}$ and $X_{M_2}$ share a boundary then the outward normal on one is the inward normal on the other, so the GHY terms cancel.

   The gravitational action including GHY and corner terms is additive, if the joints are spacelike though generally not generally for timelike joints [56]. A codimension-2 joint is spacelike if its metric is Euclidean, and timelike if it is Lorentzian; it is not determined by the metric signatures of the codimension-1 segments, i.e. $\tilde{M}$ and the $\Sigma$'s, whose shared boundary is the joint. Since additivity is only an issue for timelike joints, we only have to worry about Lorentzian bulks. There are joints between $\tilde{M}$ and the partial Cauchy slices $\Sigma_1$ and $\Sigma_2$, but since the boundaries of partial Cauchy slices are spacelike the gravitational action is additive.

   The requirement does rule out all choices of $X_M$ bulk subregions for which $X_{M_1 \cup M_2} \neq X_{M_1} \cup X_{M_2}$, such as $X_M = \Sigma_1$, if $f$ is extensive. It also rules out some functions $f$, such as non-local ones like $\int dx dy R(x) R(y)$, which will not generally give additive cost proposals.

3. Symmetry.

   $f(X_M)$ should be invariant under relabelling $\Sigma_1$ and $\Sigma_2$. An example which satisfies this requirement would be if both $\Sigma$'s are defined the same way, such as minimum volume surfaces in a Euclidean bulk. An example which does not satisfy symmetry would be if $\Sigma_1$ satisfies $K = a$ while $\Sigma_2$ satisfies $K = b$ with $a$ and $b$ different constant trace extrinsic curvatures, as then the embedding of the partial Cauchy slices will change under relabelling. It is sufficient that $X_M$ is invariant under the relabelling, though not necessary as in the trivial example $f(X_M) = 0$.

4. Covariance.

   Requiring the proposal to be covariantly defined leaves a large space of proposals. All the bulk subregions $X_M$ defined in section 3.3.1 as well as the proposals $f(X_M)$ for assigning numbers to those regions given in 3.3.2 are defined in a coordinate-independent way.

5. Non-negativity.

   We need $f(X_M) \geq 0$ for all bulks and subregions thereof to which the proposal is applicable. On the one hand it is trivial to define manifestly non-negative functions $f$. With a non-negative scalar density $\mathcal{F}$, the following is manifestly non-negative:

   $$f(X_M) = \int_{X_M} \mathcal{F}. \tag{22}$$

   Volume-type cost proposals use a constant $\mathcal{F}$. Taking the absolute value or an even power of any real-valued function makes it non-negative, so the space of non-negative $f$ is not small.

   On the other hand it can be difficult to know whether a given proposal is non-negative for its whole domain. Suppose one has a proposal for which $X_M$ is a codimension-zero region of the bulk, and $f$ is the Einstein-Hilbert action of that region, with or without boundary and corner terms. The problem is that there are on-shell asymptotically AdS spacetimes with arbitrarily negative Einstein-Hilbert action, which gives an unphysical negative cost. The action is unbounded from below and not merely negative, so non-negativity cannot be restored simply by adding a constant. Examples that demonstrate this unboundedness of the Einstein-Hilbert term can be constructed in two ways: either by making the action arbitrarily negative over a finite spacetime volume, or by making it negative (but finite) over an arbitrarily large spacetime volume. For the first kind of example, we consider a Weyl transformation of the bulk metric $G_{\mu\nu} \to e^{2\omega} G_{\mu\nu}$ with $\omega$ supported strictly inside $X_M$, so that the spacetime is still asymptotically AdS and so boundary and corner terms of the gravitational action are unaffected. For Lorentzian or Euclidean gravity the contribution to the gravitational action from the Einstein-Hilbert term *does* change under the Weyl transformation:

   $$\sqrt{G}\mathcal{R} \to e^{(d-2)\omega}\sqrt{G}\left(\mathcal{R} - 2(d-1)\nabla^2\omega - (d-2)(d-1)(\partial\omega)^2\right). \tag{23}$$

   The Einstein-Hilbert action can thus be made arbitrarily negative with a rapidly oscillating Weyl factor. To be an on-shell solution to Einstein's equation requires[6] $-\left(\frac{d}{2}-1\right)\mathcal{R} \approx T_\mu^\mu$. This means that the matter-fields involved would have to arbitrarily strongly violate the trace energy condition (TEC) $T_\mu^\mu \leq 0$. While the TEC is satisfied for simple matter models such as pressure-less dust it does not hold in all physical situations [57,58]. An example are neutron stars which are believed to be accurately described as perfect fluids with equation of state $p = \rho$, which violates the TEC [59]. However, our construction would require $T_\mu^\mu$ to become unbounded, and it is unclear to us whether this can be accomplished by any form of reasonable matter. See also [60] for a discussion of stability issues of spacetimes in which the TEC is violated. The second kind of example, where a finite negative term is integrated over an arbitrarily large volume, was essentially already constructed in [61], where it was shown that the complexity of an AdS$_3$ black hole with generic topology behind the horizon can be made arbitrarily negative by adding handles to the Einstein-Rosen bridge. This led the authors of that paper to propose a bound on the genus of bulk spacetimes.

---

[6]We neglect the cosmological constant in this discussion because when considering unboundedness of the action we are interested in the limit $|\mathcal{R}| \gg |\Lambda|$.

From our point of view, these are arguments that seem to rule out holographic proposals for cost (or complexity) that include only the Einstein-Hilbert action. This is on the grounds that within the space of all asymptotically AdS spacetimes there are those on which the proposal evaluates to a negative value, and so cannot be interpreted as cost or complexity. As a corollary of this argument, the domain of validity of the CA proposal *cannot* be all asymptotically AdS spacetimes; there are those on which the action of the WDW patch will be negative. This is not to say that the CA proposal does not give reasonable results for spacetimes such as the eternal black hole to which it was originally applied, nor that simple modifications of the proposal say by adding the matter action cannot remedy the unboundedness of the total action.

Since path integral cost upper bounds holographic state complexity, there will be additional checks coming from late-time linear growth and the switchback effect. These are specific to TFD states and hence are only secondary to the above primary requirements. In the case of observables defined using codimension-one surfaces, [22] showed that there are infinite classes of proposals which satisfy both linear growth and the switchback effect. This could mean that these conditions on complexity are not too restrictive. However, there are valid covariant proposals which violate linear growth and/or the switchback effect. In the case of linear growth, consider when $X_M$ is a codimension-one constant curvature slice (with $R = -2$) in a BTZ black hole background. The volume of these slices saturates quickly, and hence this $f(X_M)$ can be ruled out. Similar restrictions apply when $X_M$ is codimension-zero. In the next section, we give an example of a new codimension-zero complexity proposal that exhibits late time linear growth. Furthermore, since the complexity of a perturbed TFD state is expected to exhibit switchback effect, this will constrain how we choose $X_M$ and $f(X_M)$ in shockwave geometries.

This concludes our preliminary discussion of the space of holographic cost proposals. We found that non-negativity in particular is a subtle and difficult to verify requirement, and that new proposals, unless manifestly non-negative, need to be carefully checked with a skeptical eye. A natural direction to take from here is to consider proposals of increasing intricacy, and check their non-negativity case by case. Proposals where $f$ is the volume of $X_M$ are in some sense the simplest, and their non-negativity is manifest at least in Euclidean setups. In section 4.1.1 we give an argument for why the area of $\tilde{M}$ is a physically well-motivated proposal for the complexity of $U_M$, from the perspective of $T\overline{T}$ deformations. In the remainder of this paper we will look in more detail at various gravitational action-type proposals, and in particular if and when they run afoul of the non-negativity requirement.

# 4 Connecting to holographic state complexity proposals

In section 2.3 we discussed path integral optimisation and the connection to state complexity. The path-integral cost $\mathscr{C}(M)$ in general only provides an upper bound for the operator complexity $C(U_M)$, which in turn bounds the state complexity of the final state $|f\rangle = U_M |i\rangle$,

$$\mathscr{C}(M) \geq C(U_M) \geq C(|f\rangle). \tag{24}$$

In certain special cases we might expect that an optimal path integral cost gives a reasonable state complexity.

In this section we will give some illustrative examples of path integral cost proposals that reduce to existing holographic state complexity proposals. In each case we fix a proposal for cost of the bulk path integral on a bulk subregion, minimise this cost over an appropriate set of $\tilde{M}$, and show that the resulting minimal cost matches a state complexity proposal. When

optimising over path integral state preparation what we hold fixed is the final state, and we should allow for different bulk geometries as well as different subregions of each geometry. We will minimise cost with respect to $\tilde{M}$ within a fixed bulk geometry, and then argue that the cost cannot be lowered by varying the geometry, but this should be considered a simplifying feature of the particular cost proposals we are dealing with rather than a general feature. Note also that more than one path integral cost proposal can reduce to a given state complexity proposal; we give two that reduce to the CV conjecture.

## 4.1 Complexity equals volume from cost equals boundary volume

We now give a path integral cost proposal that when optimised reduces to the CV state complexity proposal. Consider any asymptotically Euclidean AdS spacetime of any dimension. We are looking to connect to state complexity, so as per the discussion from section 2.3 the appropriate set of codimension-1 surfaces $\tilde{M}$ over which we will minimise path integral cost all have fixed boundaries in common, see figure 2. We take $\Sigma_1$ to be a bulk point, so the path integrals really are preparing the state from nothing. Each $\tilde{M}$ in the set we are 'optimising' over then has one boundary, which is fixed. The cost proposal we will use is

$$\mathscr{C} = \text{Vol}[\tilde{M}]. \tag{25}$$

We just wish to show that the $\tilde{M}$ that minimises or 'optimises' this cost is the maximal volume slice in the Lorentzian continuation of the space, so we suppress constants of proportionality. The $\tilde{M}$ that miminises the path integral cost (25) is the minimal volume surface in the set with fixed boundary. We label the minimal volume surface $\tilde{M}^*$.

Naively we could leave $\Sigma_2$ unspecified because it does not play a role in the cost proposal. Can we lower the volume of $\tilde{M}^*$ and so the path integral cost by allowing the background geometry to vary? The answer is generally yes, but suppose $\Sigma_2$ is a subregion of a minimal volume slice of a given Euclidean geometry. Since $\Sigma_2$ lies on a minimal volume slice, and by definition $\partial \tilde{M} = \Sigma_2$, we have that $\tilde{M}^* = \Sigma_2$. This means that we cannot lower the volume of $\tilde{M}^*$ without changing the geometry on $\Sigma_2$, which is forbidden by the requirement of keeping the final state fixed in this optimisation procedure.

Gaussian normal coordinates adapted to $\tilde{M}^*$ are

$$ds^2 = d\tau^2 + g_{ij}(\tau, x)dx^i dx^j, \tag{26}$$

with $\tilde{M}^* : \tau(x^i) = 0$. Suppose this minimal volume surface lies on a time reflection symmetric slice $\Sigma$ of the Euclidean space, which implies that the extrinsic curvature $K_{ij}^{(\tau)}$ vanishes. This won't generally be the case since being a minimal volume surface only guarantees that the trace $K^{(\tau)} = 0$ vanishes but let us assume it. We may then analytically continue to Lorentzian signature $\tau \to it$, and second order shape variations in the direction normal to $\tilde{M}$ flip sign:

$$\frac{\delta^2}{\delta\tau(y)\delta\tau(y')}\text{Vol}[\tilde{M}] = -\frac{\delta^2}{\delta t(y)\delta t(y')}\text{Vol}[\tilde{M}], \tag{27}$$

and so $\tilde{M}^*$, which is the global minimal volume in the Euclidean space, is a local maximum in the analytic continuation to Lorentzian spacetime.

We have shown how to reduce to the CV state complexity proposal after finding the $\tilde{M}$ which minimises or 'optimises' the path integral cost (25). Our assumptions are that $\tilde{M}^*$ lies on a time reflection slice, and that the surface is the global, not just a local, maximum in volume in the Lorentzian spacetime. Note that we have only shown how to match CV conjecture at the point of time reflection symmetry.

It would have been preferable to find a cost proposal that applies to the same Lorentzian bulk as the CV conjecture, rather than the Euclidean continuation. The basic obstacle is that

our optimisation procedure involves *minimising* a cost, while the CV conjecture *maximises* a volume. We do not rule out the possibility of a Lorentzian cost proposal that reduces to the CV conjecture, but we were not able to find one that satisfies all the physical requirements.

### 4.1.1 Heuristic justification for cost proposal from $T\overline{T}$

We are taking a phenomenological approach to cost proposals, rather than try to justify them from a bottom-up gate-counting physical picture. We do however have a heuristic justification for this subsection's cost equals boundary volume proposal which we find appealing and will describe. Similar arguments have previously been made in [53, 62].

Consider a Euclidean path integral of a $T\overline{T}$-deformed theory defined on some two-dimensional manifold $M$. The deformation parameter $\lambda$ in a $T\overline{T}$-deformed theory is related to the scale of non-locality $L_{nl}$ by

$$L_{nl}^2 \sim \lambda. \tag{28}$$

One way of arguing for this relation is from the fact that the $T\overline{T}$ deformation of the free boson action is the Nambu-Goto string action, with string length $l_s^2 \sim \lambda$ [54].

We may discretize the path integral with a tensor network if we assume that each region of proper area $L_{nl}^2$ represents one tensor. Then the total number of tensors is

$$\mathscr{C}(M) \sim \int_M dx\,d\tau \frac{\sqrt{\gamma}}{L_{nl}^2} \sim \int_M dx\,d\tau \frac{\sqrt{\gamma}}{\lambda(x,\tau)}. \tag{29}$$

The state that the path integral prepares depends on the manifold, boundary conditions, and field theory action, especially through the deformation parameter $\lambda(x,\tau)$. Following section 2.1, we can now take the CFT to be holographic, with metric inherited from the induced metric on a finite cutoff surface $z = \rho(\tau)$ in Poincaré AdS$_3$:

$$\frac{1}{\rho^2}\gamma_{ij} = g_{ij}, \tag{30}$$

with

$$\sqrt{g} = \frac{1}{\rho^2}\sqrt{1 + \dot{\rho}(\tau)^2}, \tag{31}$$

and the $T\overline{T}$ relation between cutoff and deformation parameter

$$\lambda \sim G_N \rho^2. \tag{32}$$

Substituting in (29) we get an heuristic estimate for the effective number of gates in the path integral on $M$:

$$\mathscr{C}(M) \sim \frac{1}{G_N} \int_{\tilde{M}} dx\,d\tau \frac{\sqrt{1 + \dot{\rho}^2}}{\rho^2} = \frac{\text{Vol}(\tilde{M})}{G_N}. \tag{33}$$

This completes a heuristic derivation of the cost equals boundary volume proposal. Just like in the complexity=volume proposal [19], the proportionality factor in this equation will have to depend on the choice of an additional length scale, as $\mathscr{C}(M)$ has to be dimensionless. Also, notice that although the state preparation we are considering is similar to the one in [35], the holographic path integral cost derived from a $T\overline{T}$ gate counting procedure (33) is quite different from the one based on the on-shell gravitational action that was provided there,

$$I[\rho] \sim \frac{1}{G_N} \int dx\,d\tau \frac{1 + \dot{\rho}\arctan\dot{\rho}}{\rho^2}. \tag{34}$$

## 4.2 Complexity equals volume 2.0 from cost equals bulk volume

The CV2.0 proposal asserts that the spacetime volume of the boundary-anchored WDW patch represents a holographic notion of complexity in dual quantum field theories. It seems obvious to try to obtain this from the optimization of a cost functional which is simply given by the volume of $X_M$. There are however some subtleties when trying to make this precise and as we will see the CV2.0 proposal does not quite follow. Instead, we obtain something which is more like the volume of half the WDW patch.

The first issue we need to address is the choice of the initial and final slices $\Sigma_1$ and $\Sigma_2$. It is tempting to choose these to be the future and past null cones emanating from the boundaries of $\tilde{M}$, but this would lead to a problem with the additivitiy criterion for the cost function: if we combine $\tilde{M}_1$ and $\tilde{M}_2$, the future null cone attached to the future boundary of $\tilde{M}_1$ obviously does not agree with the past null cone attached to the past boundary of $\tilde{M}_2$ (which equals the future boundary of $\tilde{M}_1$). Hence $X_{M_1} \cap X_{M_2} \neq \emptyset$ and because of this additivity will generically fail. One can also choose both $\Sigma_1$ and $\Sigma_2$ to be simultaneously future or past directed null cones, but this would manifestly lead to violations of the time-reversal criterion. Moreover, it would lead to situations where $X_M$ could become the empty set in the limit where $\tilde{M}$ becomes null. In what remains we will choose $\Sigma_1$ and $\Sigma_2$ by the property that they have vanishing scalar extrinsic curvature, but the conclusions will not be substantially different for other choices of $\Sigma_1$ and $\Sigma_2$. Given this choice for $\Sigma_1$ and $\Sigma_2$, consider the candidate cost functional

$$\mathscr{C}(M) \sim \int_{N=\mathrm{Int}(\tilde{M}\cup\Sigma_1\cup\Sigma_2)} \sqrt{|G|} + \alpha \int_{\tilde{M}} \sqrt{|g|}, \tag{35}$$

where $\alpha$ is a non-negative dimensionful constant. Such simple cost functions satisfy all the properties listed in section 3.2 that we require from a good notion of a gravitational cost. A precise expression for this cost functional also requires an overall dimensionful prefactor which we did not include in (35) and which can be chosen arbitrarily.

Let us consider this cost function in the context of the situation depicted in figure 7 in the Lorentzian context, where we want the path integral to remain defined on a timelike surface. We restrict $\tilde{M}$ to be timelike in the set to be optimised over, in order to find the *unitary* time evolution operator with the lowest cost. If one then performs optimization of (35) for timelike separated initial and final state, the second term gets arbitrarily small for an almost null boundary and the latter also leads to the minimal enclosed bulk spacetime volume. If one optimizes also over time duration at fixed initial and final state, then one gets a portion of the WDW patch. Shrinking one state to a bulk point gives rise to a 'past half' of the WDW patch bounded by $\Sigma_2$. The volume of this half of the WDW patch equals to the optimum of the cost (35), which is as close as we can get to the CV2.0 proposal. We cannot change the geometry in the past domain of dependence of $\Sigma_2$ without changing the geometry on $\Sigma_2$, so having minimised the cost while working with a fixed bulk geometry to the volume of this 'half' WDW patch we cannot lower it further by varying the interior geometry without changing the final state. Rather than creating the state from a single bulk point, we could also have asked the question what the minimum cost is of the reverse process where we use a circuit to map an initial state to a single bulk point. One could perhaps think of this as a circuit which maps the state to a completely unentangled and therefore non-geometric state represented by a single bulk point. The optimal cost for this state demolition process is then given by the volume of the future half of the WDW patch where the WDW patch is cut in two pieces by $\Sigma_1$. Overall, the conclusion of this analysis could be that the CV2.0 proposal is not just computing the cost of creating the state but rather the sum of the creation and demolition cost. It would be interesting to explore this interpretation further.

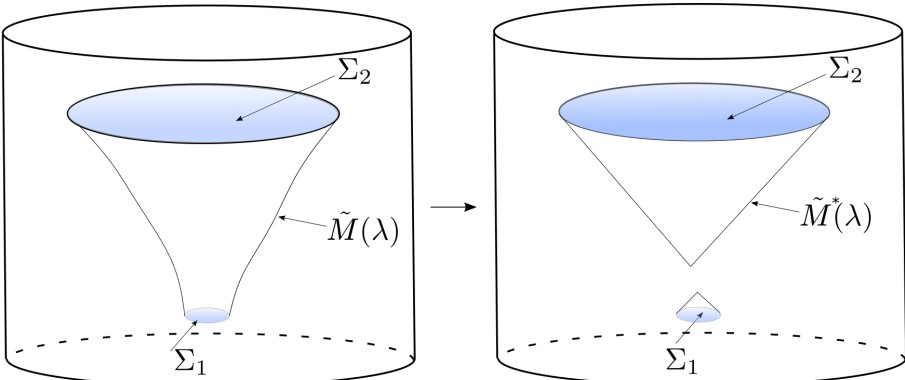

Figure 7: Optimisation of the Lorentzian cost functional given by a sum of bulk and boundary volume. The minimal cost is obtained as $\tilde{M}$ approaches an almost null surface. As $\Sigma_1$ is shrunk to a point, one obtains the past half of the WDW patch anchored to $\Sigma_2$.

### 4.3 Complexity equals volume from cost equals Euclidean gravitational action

In section 3.4 we argued that the gravitational action of codimension-0 bulk subregions is not a reasonable cost proposal because there are asymptotically AdS spacetimes, both Lorentzian and Euclidean, for which the action is negative. These actions are negative due to the conformal mode of the Ricci scalar, and are on-shell for matter configurations that violate the trace energy condition. In this section we will nonetheless use gravitational action as a cost proposal. What we have in mind is a corrected proposal that *is* non-negative: either one which excludes problematic negative action fringe cases in an ad hoc fashion, or a modification such as adding the matter action which makes the total action positive even for trace energy condition violating configurations, though we have not proven that this works. In any case we will only apply our proposal here to subregions of pure global AdS, with the matter in its vacuum configuration, so we are far from the problematic fringe cases where we would have to specify precisely how we correct our proposal to ensure positivity. We only stipulate that the presumptive correction is negligible when evaluated on pure AdS.

Cost proposals which use the gravitational action can also in some cases reduce to existing complexity proposals when optimised. In previous work we considered Euclidean Poincaré AdS$_3$ and gave a gravitational action cost proposal that reduces to the volume of the constant time slice when optimised [35]. Our cost proposal was the on-shell gravitational action of the codimension-0 bulk region bounded by two constant Euclidean Poincaré time slices and a finite cutoff radial boundary. This we claimed is dual to the cost of the Euclidean path integral in the $T\overline{T}$-deformed boundary CFT on that radial cutoff boundary. The basic idea is that we have a set of path integrals on different radial cutoff surfaces that prepare the same state, and when the gravitational action cost proposal is minimised over this set we found that it matches the CV state complexity proposal. Minimising path integral cost with respect to background geometry is inspired by the work of [26], and for Poincaré AdS$_3$ our proposal reduces to the Liouville action in agreement with their work, in the limit of a slowly varying cutoff surface. The optimum path integral maps between ground states of theories with different UV cutoffs by building up or coarse-graining away the UV structure with as little Euclidean time evolution as possible.

In this subsection we will show that when our gravitational action proposal is applied to global AdS we again match with the CV state complexity proposal. The purpose is two-fold: (1) to show that there is more than one cost proposal, in this case cost equals boundary volume and cost equals gravitational action, that can reduce to a given complexity proposal

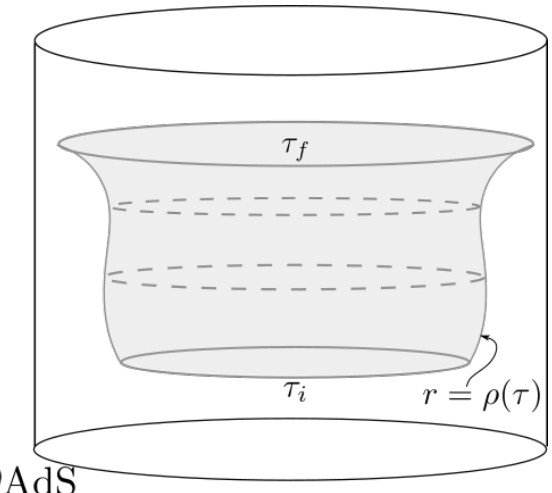

Figure 8: The subregion of Euclidean global AdS whose gravitational action we propose to be the cost of the path integral on the finite cutoff boundary.

when minimised over a suitable set of path integrals, and (2) to give further evidence that the cost proposal based on gravitational action that we gave in [35] is a reasonable one. For our gravitational action proposal we do not know whether (or necessarily expect that) it reduces to the CV conjecture in other asymptotically AdS spacetimes than the Euclidean Poincaré and global AdS examples that we have explicitly checked.

### 4.3.1 Gravitational action

Let us calculate the on-shell gravitational action between constant time slices in global AdS, with a variable finite cutoff boundary, see figure 8. Consider Euclidean AdS$_{d+1}$ with unit AdS length in global coordinates:

$$ds^2 = (1 + r^2)d\tau^2 + \frac{dr^2}{1 + r^2} + r^2 d\Omega_{d-1}^2 \,. \tag{36}$$

We assume the cutoff surface $\tilde{M}$ has spherical symmetry, $r = \rho(\tau)$. We define $\Sigma_1$ and $\Sigma_2$ to be $K = 0$ surfaces, which in this case will be just constant time $\tau = \tau_{1,2}$ slices. We will be minimising the on-shell gravitational action over cutoff boundary surfaces $r = \rho(\tau)$ with fixed initial and final cutoff $r_1 = \rho(\tau_1)$ and $r_2 = \rho(\tau_2)$. Different boundary surfaces define different path integrals which evaluate the transition amplitude between the ground states of a holographic CFT with Euclidean time dependent $T\overline{T}$ deformation. Fixing $r_1$ and $r_2$ fixes the initial and final $T\overline{T}$ deformation.

Let us calculate the on-shell gravitational action of the region depicted by figure 8. We assume the cutoff surface has spherical symmetry. Consider Euclidean AdS$_3$ with unit AdS length in global coordinates:

$$ds^2 = \frac{1}{\cos^2\theta} \left( d\tau^2 + d\theta^2 + \sin^2\theta \, d\phi^2 \right) \,. \tag{37}$$

The asymptotic boundary is at $\theta = \pi/2$. We want the on-shell gravitational action of the region $N$, which is bounded by $\tau_1 = 0$, $\tau_2 = T$, and the radial cutoff surface $\theta = \theta(\tau)$. The full gravitational action including corner terms is

$$I = \frac{1}{\kappa} \int_N d^3x \, \sqrt{G}(\mathcal{R} + 2) + \frac{2}{\kappa} \int_{\tilde{M}} d^2x \, \sqrt{g}K + I_c \,. \tag{38}$$

The extrinsic curvature of the surface $\theta = \theta(\tau)$ is

$$K = \frac{-\ddot{\theta}\tan\theta + (1 + 2\tan\theta^2)(1 + \dot{\theta}^2)}{\sec\theta\tan\theta(1 + \dot{\theta}^2)^{\frac{3}{2}}}.$$ (39)

Using this, the action takes the simple form

$$I = \frac{2}{\kappa}\int d\phi\, d\tau\left(\frac{1}{\cos^2\theta} - \frac{\ddot{\theta}\tan\theta}{(1 + \dot{\theta}^2)}\right) + I_c.$$ (40)

This can further be simplified by partially integrating over $\tau$ as

$$I = \frac{2}{\kappa}\int d\phi\, d\tau\left(\frac{1}{\cos^2\theta} + \frac{\dot{\theta}\arctan\dot{\theta}}{\cos^2\theta}\right) - \frac{2}{\kappa}\int d\phi\, \tan\theta\arctan\dot{\theta} + I_c.$$ (41)

If the boundary is not smooth at a corner situated at $\tau = \tau_c$, then the action receives an additional contribution given by the Hayward term

$$I_c = \frac{2}{\kappa}\int d\phi\, \tan\theta(\tau_c)(\arctan\dot{\theta}(\tau_c) + \pi/2).$$ (42)

Including this term from a single corner, the total action now is

$$I = \frac{2}{\kappa}\int d\phi\, d\tau\left(\frac{1 + \dot{\theta}\arctan\dot{\theta}}{\cos^2\theta}\right) + \frac{2\pi^2\tan\theta(\tau_c)}{\kappa}.$$ (43)

This result is closely related to our previous result for Euclidean Poincaré AdS$_3$ [35]. Varying this action allows us to find surfaces that extremise the gravitational action. The equations of motion are

$$\frac{\ddot{\theta} - \tan\theta(1 + \dot{\theta}^2)}{\cos^2\theta(1 + \dot{\theta}^2)^2} = 0.$$ (44)

Before solving the above equation, we see that there will always be a solution when $|\dot{\theta}| \to \infty$. In this limit the surface turns in to a equal-time slice. The most general solution for $\theta(\tau)$ is given by

$$\theta(\tau) = \arcsin\left(\alpha\sinh(\tau) + \beta\cosh(\tau)\right).$$ (45)

As expected from the discussion in [35], these $\theta(\tau)$ describe surfaces of constant scalar curvature $R = -2$. The circuits whose boundary surface is given by the above $\theta(\tau)$ or in terms of $r(\tau) = \tan\theta(\tau)$ extremise the action (43) and hence the cost of the circuit. More specifically, consider the circuit preparing the ground state $|0\rangle_{\theta_f}$ at some cut-off $\theta_f$ starting from a trivial initial state. Such a circuit $\theta(\tau)$, running from $\tau = 0$ to $\tau = T > 0$ is given by (45) with $\theta(0) = 0$ and $\theta(T) = \theta_f$. The cost can now be calculated from the value of the on-shell action, and is given by

$$I = \frac{4\pi}{\kappa}\left(T + \arctan\left(\frac{\tan\theta_f}{\tanh T}\right)\tan\theta_f\right) + \frac{2\pi^2\tan\theta_f}{\kappa}.$$ (46)

Minimum value of this optimised cost for preparing $|0\rangle_{\theta_f}$ is achieved for $T = 0$, when the surface is a constant time slice. The minimum value is

$$I_{min} = \frac{4\pi^2\tan\theta_f}{\kappa}.$$ (47)

The volume of the constant time slice with a radial cut-off at $\theta_f$, Vol($\theta_f$) is equal to $2\pi \tan\theta_f \tan\frac{\theta_f}{2}$. Using this, and $\frac{1}{\kappa} = \frac{c}{24\pi}$ we can rewrite the above value as

$$I_{min} = \frac{c}{12} \frac{\text{Vol}(\theta_f)}{\tan\frac{\theta_f}{2}}. \tag{48}$$

The minimum cost is indeed proportional to the volume of the constant time-slice. As the radial cut-off is taken to infinity, $\theta_f \to \pi/2$, we see that the proportionality constant is exactly $\frac{c}{12}$.

We should again ask whether the bulk path integral cost can be lowered by allowing the background geometry to vary. In a similar resolution to the previous subsection we simply specify $\Sigma_2$ to lie on the constant time slice. Then the cost-minimising $\tilde{M}$ and $\Sigma_2$ coincide and it is not possible to lower the cost without changing the final state.

### 4.3.2 Kinematic space analysis

The on-shell gravitational action for global AdS$_3$ with a time-dependent boundary surface $\theta(\tau)$, given in (40), can be rewritten as

$$\begin{aligned}
I &= \frac{2}{\kappa} \int d\phi \, d\tau \left( \frac{1}{\cos^2\theta} - \frac{\ddot{\theta}\tan\theta}{(1+\dot{\theta}^2)} \right) + I_c \\
&= \frac{2}{\kappa} \int d\phi \, d\tau \left( \tan\theta \left( \frac{\tan\theta(1+\dot{\theta}^2) - \ddot{\theta}}{(1+\dot{\theta}^2)} \right) + 1 \right) + I_c.
\end{aligned} \tag{49}$$

Ignoring an overall additive factor and the corner term, the remaining action reads

$$I = \frac{2}{\kappa} \int d\phi \, d\tau \, \tan\theta \left( \frac{\tan\theta(1+\dot{\theta}^2) - \ddot{\theta}}{(1+\dot{\theta}^2)} \right). \tag{50}$$

This action can in fact be reproduced by considering the kinematic space of bulk curves. The data of time-dependent bulk surfaces $\theta(\tau)$ can be encoded in the boundary. This is done by giving a pair of boundary points $(\tau_1(\tau), \tau_2(\tau))$ such that the bulk geodesic starting at $\tau_1(\tau)$ and ending at $\tau_2(\tau)$ is tangent to the bulk surface at the point $(\tau, \theta(\tau), 0)$. Such pairs of points form the kinematic space, with a metric fixed by conformal invariance

$$ds_{ks}^2 = -\frac{4d\tau_1 d\tau_2}{\sinh^2(\tau_1 - \tau_2)}. \tag{51}$$

The explicit dependence of the boundary points $\tau_{1,2}(\tau)$ on the bulk time $\tau$ is

$$\tau_{1,2}(\tau) = \tau + \log\left( \frac{1 \pm \cos\theta \sqrt{1+\dot{\theta}^2}}{\sin\theta + \dot{\theta}\cos\theta} \right). \tag{52}$$

Now consider an action built using kinematic space, as

$$S_{ks} = \frac{2}{\kappa} \int (\tan\theta \, d\phi) \, ds_{ks}. \tag{53}$$

Using (52) and the distance in kinematic space $ds_{kin}$ the above action equals

$$S_{ks} = \frac{2}{\kappa} \int (\tan\theta \, d\phi) \, ds_{ks} = \frac{2}{\kappa} \int d\phi \, d\tau \, \tan\theta \left( \frac{\tan\theta(1+\dot{\theta}^2) - \ddot{\theta}}{(1+\dot{\theta}^2)} \right). \tag{54}$$

This matches exactly to the on-shell gravitational action obtained above after ignoring an overall additive factor and the corner term, and is the global AdS equivalent to the result for Poincaré AdS that we had obtained in section 4.2 of [35].

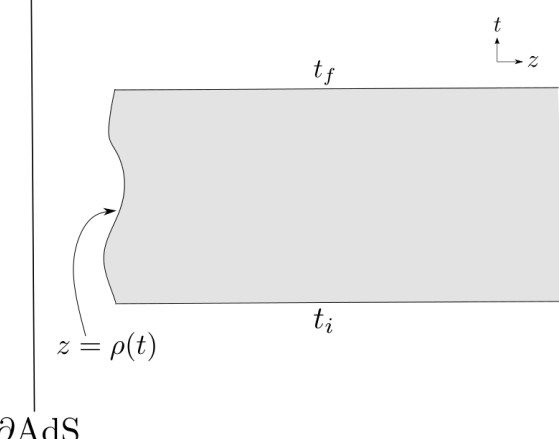

Figure 9: A subregion of Lorentzian Poincaré AdS between two constant time slices with a time dependent boundary. The path integral of the $T\overline{T}$-deformed theory on the $z = \rho(t)$ boundary defines a path through the space of unitaries, and the gravitational action of the shaded region, including boundary and corner terms, is the natural Lorentzian extension of our previous proposal [35] for the length or cost of that path.

### 4.4 Obstacles to obtain the complexity equals action proposal from a cost

Here we will see that the naive extension of our Euclidean cost proposal from [35] to Lorentzian Poincaré AdS$_3$ gives unphysical results. The proposal is to extremise the action of the subregion of Lorentzian Poincaré AdS$_3$,

$$ds^2 = \frac{-dt^2 + dz^2 + dx^2}{z^2}, \tag{55}$$

shown in figure 9 to see whether these extrema can be sensibly interpreted as circuit cost or complexity of Lorentzian time evolution between the initial and final time slice. In close analogy to the Euclidean result given in equation (2.10) of [35], the gravitational action of this subregion is (see appendix A for details of the calculation)

$$I = \frac{1}{8\pi G_N} \int dx \int_{t_i}^{t_f} dt \left( \frac{1 - \dot\rho \operatorname{arctanh} \dot\rho}{\rho^2} \right). \tag{56}$$

This action is unbounded from below; it can be seen that $I \to -\infty$ in the limit of the cut-off surface becoming null, $|\dot\rho| \to 1$. Whether the action is bounded from above depends on the boundary conditions. The solution to the Euler-Lagrange equations for general boundary conditions $\rho(t_i) = \rho_i$, $\rho(t_f) = \rho_f$ is

$$\rho(t) = \sqrt{t^2 + At + B}, \tag{57}$$

where

$$A = -\frac{t_f^2 - t_i^2 + \rho_i^2 - \rho_f^2}{t_f - t_i}, \quad B = \frac{(t_f - t_i)t_i t_f + t_f \rho_i^2 - t_i \rho_f^2}{t_f - t_i}. \tag{58}$$

This solution is a local maximum of (56). In contrast to what we observed in the Euclidean case in [35], this timelike cutoff surface bends outwards from $\rho_{i/f}$ towards the asymptotic boundary, as it tries to maximise the $\rho^{-2}$ factor. The solution fails to be real when the time interval becomes too large,

$$t_f - t_i > \rho_i + \rho_f. \tag{59}$$

Roughly speaking this is because when the time interval is large compared to the spatial initial and final cutoffs, the cutoff surface can get to the asymptotic boundary and back without $\dot{\rho}$ becoming large enough to flip the sign of numerator in (56). Once at the asymptotic boundary, the denominator diverges, leading to an action that is unbounded from above, which is why we find no (real) solution that extremises the action. In fact, the action can be arbitrarily negative for other cutoff surfaces as well. For a more generic cutoff surface $\rho(t, x) = \sqrt{r(x)^2 + t^2}$ (this is derived as a solution to equation (71) which we discuss in section 5), the action is given by

$$I = \frac{2}{\kappa} \int dt\,dx\, \frac{r(x)r''(x)}{(t^2 + r(x)^2)(1 + r'(x)^2)}\,. \tag{60}$$

We see that the above integral can turn out to be negative depending on the choice of $r(x)$. For example, with $r(x) = \sin(\omega x)$ the action computed in the region $t, x \in [0, 1]$ is negative and proportional to $\omega^2$. Thus, by making the cutoff surface more wavy the action can be made arbitrarily negative.

## 4.5 Linear growth at late times for BTZ black hole

The notion of holographic complexity was initially used for describing the growth of black hole interiors for long times. Any measure of complexity must exhibit a late time linear growth in black hole backgrounds. We saw that the cost function given by Euclidean gravitational action in the region bounded by two $K = 0$ slices $\Sigma_{1,2}$ and $\tilde{M}$ was well-defined and gave sensible results in global $AdS$. Moreover, choosing a trivial $\Sigma_1$ and optimising over $\tilde{M}$ lead to the action between a constant scalar curvature slice and $\Sigma_2$. In this subsection, we will assume we can evaluate the Lorentzian action between these surfaces in the BTZ black hole and verify that this exhibits linear growth at late times. Therefore, it is a candidate new holographic complexity proposal.

Consider a BTZ black hole with horizon radius $r_h = 1$ and AdS radius $L = 1$. Then using Kruskal coordinates, the metric takes the simple form [63]

$$ds^2 = -\frac{4\,dU\,dV}{(1 + UV)^2} + \frac{(1 - UV)^2}{(1 + UV)^2}d\phi^2\,. \tag{61}$$

The mass of the black hole is $M = \frac{r_h^2}{8GL^2} = \frac{1}{8G}$. We have the asymptotic AdS boundaries located at $UV = -1$ and the horizons at $UV = 0$. Maximal volume slices have vanishing trace of extrinsic curvature

$$K = \frac{(U^2 V^2 - 1)U'' + 2(UV - 3)U'(U - U'V)}{4(UV - 1)|U'|^{3/2}} = 0\,. \tag{62}$$

Here, $'$ denotes a $V$ derivative. In fact, the maximal surfaces are best described in Eddington-Finkelstein coordinates

$$ds^2 = -f(r)dv^2 + 2dv\,dr + r^2 d\phi^2\,, \tag{63}$$

with $f(r) = r^2 - 1$. Then, the shape $v(r)$ of the maximal surfaces is given as [64, 65]

$$\frac{dv}{dr} = \frac{\sqrt{f(r)r^2 + c^2} - c}{f(r)\sqrt{f(r)r^2 + c^2}}\,. \tag{64}$$

The constant $c$ determines the boundary time at which the maximal surface is anchored. It goes from $c = 0$ for the surface anchored at $t_L = t_R = 0$ to $c = \frac{1}{2}$ for the final slice. The final maximal slice for the BTZ black hole is at $r = \frac{1}{\sqrt{2}}$ and at late times maximal surfaces pile up

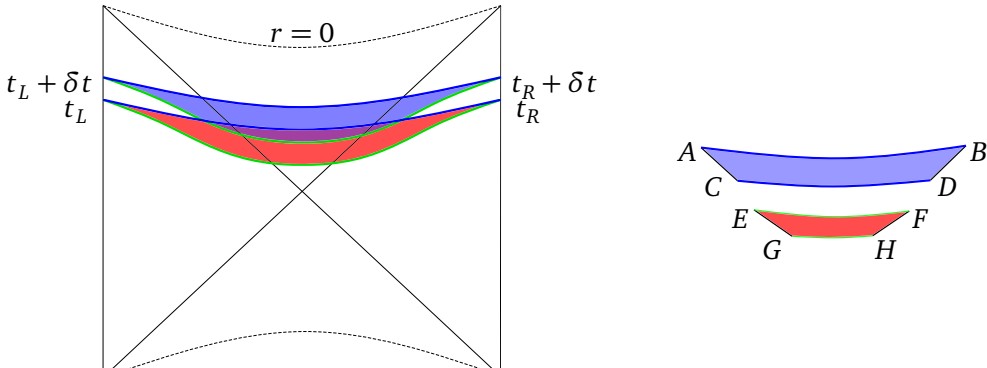

Figure 10: Growth of action in the BTZ black hole between constant curvature surfaces (blue) and maximal volume surfaces (green).

very close to this surface. Constant scalar curvature slices satisfying $R + 2 = 0$ are much easier to find, and are given by

$$UV + \lambda U + \mu V - 1 = 0 \,. \tag{65}$$

Here $\lambda$ and $\mu$ again determine where these surfaces are anchored at the boundary.

Now consider the gravitational action within the region bounded by the maximal surface and a constant curvature surface both fixed at the same boundary time $t_L = t_R = t$. To compute the growth rate, consider two such regions separated by a small boundary time $\delta t$. Let us compute the difference in the actions of these nearby regions. The total action outside the horizon is time-independent and can be ignored. This leaves us with two regions inside the horizon: action in the blue region between constant curvature surfaces minus action in the red region between maximal surfaces in figure 10. For the blue region inside the horizon, the bulk action is given by

$$\delta I_{bulk} = -\frac{1}{2G_N}\left(\tanh t + \frac{t}{\cosh^2 t}\right)\delta t \,. \tag{66}$$

To this, we also need to add boundary and corner contributions. For the boundary terms, we have the usual GHY surface terms along $AB$ and $CD$ (see figure 10), and also null boundaries along the horizon segments $AC$ and $BD$. We choose the normals to the null surface to be affinely parameterised, hence the latter terms can be set to zero. As the expansion parameter along these (Killing) horizon segments vanishes, we can likewise ignore the counter terms proposed for null surfaces [55]. The boundary terms $\delta I_{bdy}$ from the segments $AB$ and $CD$ exactly cancel the above bulk term, leaving us with four corner terms. Since these corners arise from the intersection of a spacelike surface and the null horizon, the appropriate action [66] is

$$I_{corner} = \frac{1}{8\pi G_N}\int_{\partial\Sigma} d\phi\,\sqrt{\sigma}a \,, \tag{67}$$

where $\sigma$ is the metric on the corner $\partial\Sigma$ and $a = \pm\log(\mathbf{k}\cdot\mathbf{n})$, with $\mathbf{k}$ and $\mathbf{n}$ being the normals to both the surfaces at the corner. Calculating this for corners on both sides gives

$$\delta I_{corner} = \frac{1}{2G_N}\tanh t\,\delta t \,. \tag{68}$$

For the red region, we only have the bulk and corner terms, since $K = 0$ along the boundaries. We shall argue that both these terms can be safely ignored in the late time limit. The bulk

action can be computed numerically from the shape of surface in (64). At late boundary time, since these surfaces pile up close to the final surface, this action is negligible. Computing the corner term from (67) gives

$$\delta I_{corner} = \frac{1}{2G_N c(t)} \frac{dc(t)}{dt} \delta t, \tag{69}$$

where $c(t)$ is implicitly given by (64). Again, at late boundary times $t \gg 1$, we have $c \approx \frac{1}{2}$, hence this term doesn't contribute as well. Now, adding up all the contributions we have

$$\frac{dI_{total}}{dt_{bdy}} = \frac{\tanh t}{4G_N} = 2M \tanh \frac{t_{bdy}}{2}, \tag{70}$$

where $t_{bdy} = t_L + t_R = 2t$ is the total boundary time. Since at late boundary times $\tanh \frac{t_{bdy}}{2} \approx 1$, the action grows linearly.

# 5 General methods for gravitational action proposals

Let us quickly summarize some of the achievements of the preceeding section. Following [35], we showed in section 4.3 how a cost-proposal based on the bulk gravitational action can reproduce the complexity=volume proposal on a Euclidean global AdS background (see also table 1), and we discussed how this result can be connected to the geometry of kinematic space, i.e. the space of spacelike bulk geodesics, in section 4.3.2. While there are problems with the generalisation of this ansatz to Lorentzian cases as discussed in section 4.4, it is interesting to note the prominent role that surfaces of a constant intrinsic curvature (such as (45) and (57)) play in all these attempts as solutions to the equations of motion derived by extremising the action. This motivated us in section 4.5 to propose a new complexity proposal based partially on constant intrinsic curvature surfaces that was shown to pass at least one important plausibility check, namely late time linear growth in a black hole background.

For this reason, in this section we will now give a more general analysis of the general equations derived in [35] (of which sections 4.3 and 4.4 only provide special examples). As we are about to explain, a quite generic solution method can be formulated based on foliating surfaces by geodesic curves, which in turn might suggest a deeper and more general connection to the physics and geometry of the kinematic space than what we discussed in [35] and section 4.3.2. However, a more detailed study of such a possible deeper connection will be left for future research.

## 5.1 Equations of motion

In [35] we essentially analysed a problem where co-dimension one hypersurfaces $\tilde{M}$ were embedded into AdS$_3$ according to the equation[7]

$$K_m^n K_n^m - K^2 = 0. \tag{71}$$

Herein $K_{mn}$ is the extrinsic curvature tensor of the surface and $K = K_n^n$ is its trace. Latin indices are raised and lowered with the induced metric $g_{mn}$. The potential physical interpretations of

---

[7]We follow here the notation of section 3 of [67], where latin indices refer to the induced geometry of the hypersurface $\tilde{M}$ with coordinates $y^a$, greek indices refer to the ambient (bulk) spacetime $N$ with coordinates $x^\alpha$, and we can define the projector $e_a^\alpha = \partial x^\alpha / \partial y^a$. To avoid confusion concerning e.g. the Ricci scalar, we use $\mathcal{R}$ for curvature tensors of the bulk, and $R$ for curvature tensors of the induced metric. The bulk metric as throughout the paper is $G_{\mu\nu}$ while the induced metric is $g_{mn}$.

this equation are manifold. Our main interpretation in [35] was that when deriving a notion of state complexity by extremising the action of a bulk region bounded by initial and final time slices as well as a variable boundary surface, (71) arises as the equation of motion of that surface. Additionally, in section 3 of [35] we pointed out how surfaces satisfying (71) arise from flow equations which describe movement of the cutoff surface in a fixed background, while in section 4 of [35] we pointed out a connection with kinematic space. In the following, we will continue to explore these possible interpretations in more generality than what was possible in [35].

To do so, we should first point out that the derivation given in section 3.1 of [35] is independent of the number of dimensions and equally applicable to the Lorentzian case, hence from now on we take equation (71) to be the equation of interest even in the general case.[8] Also, due to the Hamiltonian constraint[9]

$$0 \equiv H = R - 2\Lambda - \left( K_m^n K_n^m - K^2 \right) , \tag{72}$$

equation (71) corresponds to demanding that the Ricci curvature $R$ of the induced metric of the surface is constant. Specifically, if we focus on three bulk dimensions and set the AdS-radius to $L = 1 \leftrightarrow \Lambda = -1$, then $\mathcal{R} = -6$ and $R = -2$. Of course, the problem of constant curvature surfaces embedded into maximally symmetric ambient spaces is well studied in the mathematical literature, see e.g. [68–75] and references therein for interesting results. However due to differences in notation and nomenclature in the mathematical literature, in the following sections we will spell out the most relevant facts for our case in our own language and try to give them a physical interpretation from the perspective of holography. In fact, as pointed out in [74], the French mathematician G. Darboux remarked more than 130 years ago that *it can be said that the total curvature has more importance in Geometry; as it depends only on the line element, it comes into play in all questions concerning the deformation of surfaces. In mathematical physics, on the contrary, it is the mean curvature [i.e. extrinsic curvature] which seems to play the dominant role* [76]. While much has happened in the world of mathematical physics since this statement was made, especially with the introduction of general relativity, at least in the AdS/CFT correspondence it still seems to hold true to this day. Namely, it is extremal surfaces (i.e. those with *vanishing* mean curvature) that play a role in the Ryu-Takayanagi formula [5], the holographic description of Wilson loops [77], or the complexity=volume proposal [19]. In this sense, our previous paper [35] as well as this one stand out as they point towards a physical role of constant Gauss curvature surfaces in the holographic dictionary.

The first observation we can make about (71) is that it can be written solely in terms of the object $K_m^n$. This is reminiscent of the paper [78], where the authors studied (one-dimensional) curves with more complicated equations of motion than merely geodesic equations. The authors there found that in some cases, it was possible to phrase these equations in terms of extrinsic curvature as a function of an affine parameter. Then, a solution can be obtained in a two-step procedure: first by solving the equation for the extrinsic curvature, and then finding an embedding for a curve that actually has this extrinsic curvature as a function of the affine parameter. Similarly, we could try to solve (71) by firstly finding any tensor $K_m^n$ (dependent on generic induced coordinates $y^a$) that satisfies this equation,[10] and then solving for the embedding of a hypersurface in the ambient spacetime that, for the correct choice of induced coordinate system, has the extrinsic curvature found in the first step of the solution procedure.

---

[8]Specifically, in equation (3.7) of [35] (where $d$ stands for the dimension of the entire bulk spacetime), plugging in the relation $\pi_{mn} = -(K_{mn} - K g_{mn})$ shows that (71) arises as the flow eqation.

[9]Here and for the rest of the paper, we only consider vacuum spacetimes in the bulk.

[10]Because (71) does not contain derivatives, this reduces to a pointwise matrix equation. Any section dependent on parameters $y^a$ in the space of matrices that satisfy the constraint (71) would then be a valid solution of the first step of this procedure.

Unfortunately we have not been able to carry out this procedure in general, hence in the next subsection we will study a particular ansatz to solve (71).

## 5.2 Solution method, totally geodesic foliations

It is trivial to see that an ansatz of the form

$$K_{mn} = m_m m_n k, \tag{73}$$

with some vector $m$ and some function $k$ will automatically satisfy (71). We can demand $m$ to be normalized, or alternatively we could allow $m$ to be unnormalised and absorb $k$ into its norm *up to* an overall sign. Which convention is more useful depends on the problem at hand. Firstly, let us discuss how general this ansatz is. For two dimensional surfaces, (71) is equivalent to $\det K_{mn} = 0$ and hence to (73), i.e. this ansatz is generic in this case. For higher dimensions however, (73) only covers a small subset of the solutions of (71).

For a hypersurface embedded into an ambient spacetime, we can utilize the Codazzi equations. Besides (71), a consistent embedding into an ambient space with given $K_{mn}$ needs to satisfy the following equations [67]:

$$\mathcal{R}_{\alpha\beta\gamma\delta}e_a^\alpha e_b^\beta e_c^\gamma e_d^\delta = R_{abcd} \pm (K_{ad}K_{bc} - K_{ac}K_{bd}), \tag{74}$$

$$\mathcal{R}_{\mu\beta\gamma\delta}n^\mu e_b^\beta e_c^\gamma e_d^\delta = K_{bc|d} - K_{bd|c}, \tag{75}$$

$$\left(\mathcal{R}_{\alpha\beta} - \frac{1}{2}\mathcal{R}G_{\alpha\beta}\right)n^\beta e_a^\alpha = K_{a|b}^b - K_{,a}. \tag{76}$$

Note that the bracket in (74) automatically vanishes with our ansatz, hence if the ambient space has a Riemann-tensor of the form of a maximally symmetric spacetime

$$\mathcal{R}_{\alpha\beta\gamma\delta} = \frac{\mathcal{R}}{d(d-1)}\left(G_{\alpha\gamma}G_{\beta\delta} - G_{\alpha\delta}G_{\beta\gamma}\right), \tag{77}$$

then due to the projections in (74) the Riemann tensor of the induced metric will have a similar maximally symmetric form in terms of the induced metric and its Ricci scalar. Hence our ansatz (73) necessarily describes a hypersurface whose induced metric is (locally[11]) maximally symmetric, and because in an $\text{AdS}_d$ background (71) implies a negative induced curvature, the induced metric has to be locally $\text{AdS}_{d-1}$. Furthermore, under the assumption of embedding into a locally AdS space, the left-hand sides of (75) and (76) will vanish because $G_{\alpha\beta}n^\beta e_a^\alpha = 0$, giving us an interesting set of differential equations for $m$ and $k$. Assuming we can set $k = \pm 1$ at least in certain regions of the hypersurface, (75) gives:

$$0 = m_c \nabla_d m_b + m_b \nabla_d m_c - m_d \nabla_c m_b - m_b \nabla_c m_d. \tag{78}$$

So in general, we would have to find a vector field $m$ in a locally AdS space that satisfies (78), and then see whether there actually is a surface embedded into AdS that has the corresponding extrinsic curvature and induced metric in the induced coordinate system of our choice.

Let us now focus on $d = 3$ dimensional ambient spaces, i.e. two dimensional hypersurfaces for the moment. Clearly, $m^a$ is a vector in the tangent space to the hypersurface, so there is one perpendicular direction in the tangent space, and we introduce the tangent vector field $l^a$, such that $l^a m_a = 0, l^a l_a = const$. What can we learn about this vector field? Take equation (78), and contract it with $l^c l^b$:

$$0 = m_d l^c l^b \nabla_c m_b \Rightarrow 0 = l^c l^b \nabla_c m_b. \tag{79}$$

---

[11]BTZ black holes [63] are examples for spaces which are locally AdS, but have interesting global properties.

We hence know

$$l^b m_b = 0 \Rightarrow 0 = l^c \nabla_c (l^b m_b) = (l^c \nabla_c l^b) m_b + \underbrace{l^c l^b \nabla_c m_b}_{=0} \,. \tag{80}$$

That means the projection of the vector $l^c \nabla_c l^b$ on the $m_b$ direction has to vanish. As we assume the hypersurface worldvolume to be 2-dimensional, the only other direction is $l_b$. We find:

$$(l^c \nabla_c l^b) l_b \propto l^c \nabla_c l^b l_b = 0 \,. \tag{81}$$

It follows that (80) and (81) together imply the geodesic equation

$$l^c \nabla_c l^b = 0 \,, \tag{82}$$

in the induced metric of the hypersurface. Thus our ansatz (73) implies that the integral lines of the normalised vector-field perpendicular to the direction $m_a$ have to be geodesics which foliate the hypersurface. So far, we are explicitly talking about the geodesic equation with respect to the induced metric, but as (73) implies $K_{ab} l^a l^b = 0$ these curves also have vanishing extrinsic curvature in the normal direction to the hypersurface. Hence these curves foliating the hypersurface will also be geodesics with respect to the ambient metric. We can show this explicitly. The relation between the covariant derivative in the ambient space $X_{;\beta}$ and the covariant derivative in the induced metric $X_{|b}$ gives [67]

$$l^\alpha_{;\beta} e^\beta_b = l^a_{|b} e^\alpha_a \pm l^a K_{ab} n^\alpha \,. \tag{83}$$

Contracting (83) with $l^b$, we find

$$\underbrace{l^\beta l^\alpha_{;\beta}}_{\text{ambient space geodesic eq.}} = \underbrace{l^b l^a_{|b}}_{\text{induced metric geodesic eq.}} e^\alpha_a \pm l^b l^a K_{ab} n^\alpha \,. \tag{84}$$

Herein, $l^\beta$ is the ambient space form of the vector field $l^b$ in the hypersurface. Thus, if the 2d hypersurface is foliated by curves (with tangent vector $l$) that are both geodesics of the ambient space and the induced metric (i.e. totally geodesic), then necessarily $K_{ab} l^a l^b = 0$. On the other hand, if $K_{ab} l^a l^b = 0$ is given and as derived above the geodesic equation with respect to the induced metric is satisfied, then so will be the geodesic equation with respec to the ambient metric.

To summarise, we have shown that in AdS$_3$, the constant curvature surfaces that we are trying to find as solutions of (71) (which implies (73) in three bulk dimensions) are foliated by curves that are geodesics both with respect to the ambient space and the induced metric. This is just the AdS equivalent of the well known statement in $\mathbb{R}^3$ that all developable surfaces (i.e. $R = 0$) are ruled surfaces (i.e. foliated by straight lines in $\mathbb{R}^3$) [75], however both in this and in our case, the converse is not true. We can use this realisation to construct hypersurfaces that will solve (71) subject to quite generic boundary conditions, as we demonstrate in section 5.3. Interestingly, the result of this section hence implies a relation between solutions of (71) in three bulk dimensions and the abstract space of geodesics of the bulk spacetime. This space of geodesics generalises the well known kinematic space [79] which we use for example in section 4.3.2 by including geodesics not restricted to an equal time slice as well as timelike geodesics [80–82]. In three bulk dimensions, this space will be four dimensional, and as we have shown in this section, a surface solving (71) will correspond to a curve in this space of geodesics, each point along this curve corresponding to one geodesic which constitutes a slice of the codimension-one surface $\tilde{M}$ in the bulk. While there is a considerable freedom of how such curves in the space of geodesics can look like, corresponding in part to our freedom

of choosing arbitrary boundary conditions for the surface $\tilde{M}$ in the bulk, not *every* such curve generates a bulk surface that solves (71). It would hence be interesting to try and rephrase equation (71) as a constraint on curves in the space of bulk geodesics, but we leave this for future work. Furthermore, it was discussed in [82] that the space of timelike geodesics in AdS$_3$ can be mapped to the space of coherent states of the CFT. Under this identification, the Lorentzian solutions which we will later construct in section 5.5 would receive the interpretation of corresponding to (closed) paths in this space of states, however we will also leave it to future research to investigate the possible significance of this observation.

## 5.3 Examples

In [35], we derived solutions to (71) in Euclidean Poincaré AdS$_3$ anchored to two constant time slices at different times on the boundary. The solution was a translation invariant hypersurface with semi-circular cross sections, and we remarked that these semicircular cross-sections are geodesics of the ambient space. In light of the results discussed in the previous subsection, this observation is now not surprising anymore. In fact, we can quite easily generalise our solution to the case where translation invariance is broken, assuming only a mirror symmetry between initial and final time slice. This is done by the ansatz

$$z(t, x) = \sqrt{r(t)^2 - x^2},\tag{85}$$

where $r(t)$ is an arbitrarily varying (half) width along the $t$-axis, and we have used the usual coordinate system on (Euclidean) Poincaré AdS$_3$ that gives us the line element

$$ds^2 = \frac{1}{z^2}\left(dt^2 + dx^2 + dz^2\right),\tag{86}$$

where from now on we set the AdS-scale to $L = 1$ for simplicity. The case in [35] was simply $r(t) = const$. This embedding indeed satisfies (71). Likewise, in Lorentzian global AdS$_3$ with line element

$$ds^2 = \frac{1}{\cos(\theta)^2}\left(-dt^2 + d\theta^2 + \sin(\theta)^2 d\phi^2\right),\tag{87}$$

we can now easily construct the surface

$$t(\phi, \theta) = t_{bdy}\left[\arctan\left(\sqrt{\csc^2(\theta)\sec^2(\phi) - 1}\right)\right],\tag{88}$$

which can be verified to satisfy (71), and where $t_{bdy}[\phi]$ is the boundary condition at the asymptotic boundary $\theta = \pi/2$ which we assume to be symmetric under $\phi \rightarrow -\phi$. See figure 11 for examples.

The ease with which we can now construct solutions to (71) allows us to directly settle certain interesting physical questions, such as those concerning uniqueness of solutions. Consider again Euclidean Poincaré AdS$_3$, and on the boundary we want our hypersurface to be anchored on an ellipse in the $t - x-$plane with semi minor axis $= 1$ along the $t$-axis and semi major axis $= 2$ along the $x$-axis. Interestingly, we can construct hypersurface embeddings similar to (85) in two ways: with a foliation in terms of semi-circular arcs parallel to the $t$-axis, or with a foliation in terms of semi-circular arcs parallel to the $x$-axis, see figure 12. Both these embeddings satisfy (71), but one reaches farther into the bulk than the other. Hence for given boundary conditions, solutions to (71) will generally not be unique.

We will make more use of this solution generating method in subsection 5.5, but before that we will comment on the importance of the Gauss-Bonnet theorem in our context.

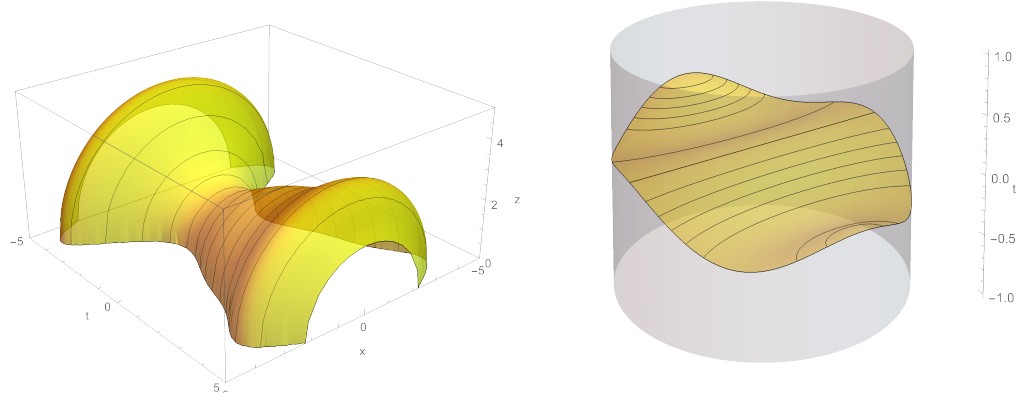

Figure 11: Left: Example of (85) for $r(t) = 3 + \frac{1}{2}\sin(t) - \cos(t^2/5)$. This satisfies (71). For this solution the sign of $K$ switches in between points on the surface, hence $k$ in (73) can not be globally absorbed into the normalisation of $m$. Note we are working in the Euclidean case, so the norm of $m$ has to be positive. Right: Example of (88) for $t_{bdy}[\phi] = \frac{1}{8}\cos(4\phi) - \frac{\cos(\phi)}{4}$. The asymptotic boundary of global AdS is depicted as a grey cylinder.

## 5.4 Implications of the Gauss-Bonnet theorem

As we are searching for surfaces of constant scalar curvature, in the case of two-dimensional surfaces $\tilde{M}$ it is quite natural to consider the implications of the Gauss-Bonnet theorem

$$\int_{\tilde{M}} \frac{R}{2} dV + \int_{\partial\tilde{M}} k_g ds + \sum_{\text{corners } c} \alpha_c + \sum_{\text{conical sing. } s} \beta_s = 2\pi\chi \,, \tag{89}$$

see e.g. [83]. Herein, the first term is an integral of the Gaussian curvature over the volume of the surface. The second term is an integral over the geodesic curvature along the boundary lines of the manifold. The third term takes into account contributions from corners in these boundaries. Here, $\alpha_c$ is the external angle at every corner by which the boundary changes direction, i.e. $\pi$ minus the interiour angle at the corner. This angle has to be defined with a positive sign at convex corners and a negative sign at concave corners. Lastly, the fourth term (see [84]) takes into account contributions from conical singularities in the manifold $\tilde{M}$, where $\beta_s$ is the conical deficit angle. These specific terms are rarely mentioned in descriptions of the Gauss-Bonnet theorem, but they will be especially important in our context, and so we will explain them in more detail in appendix B. On the right hand side of the equation, $\chi$ is the Euler characteristic. Importantly, the theorem is valid both in the Euclidean and Lorentzian case, however, in the latter angles have to be replaced by Lorentzian analogues and $\chi \equiv 0$, see [85–88] and the discussion in appendix B.

As we are concerned with constant curvature surfaces, the first term in (89) can be simplified to the product $RV/2$ where $V$ is the total volume of the surface. Let us for the moment assume smooth surfaces, without corners or conical singularities. We can hence write:

$$\frac{RV}{2} = 2\pi\chi - \int_{\partial\tilde{M}} k_g ds \Rightarrow V = \frac{2\int_{\partial\tilde{M}} k_g ds - 4\pi\chi}{-R} \,. \tag{90}$$

Hence, because we fix the value of $R$, there is (for fixed topology) a direct relation between volume $V$ and geodesic curvature of the edge of the surface $\int_{\partial\tilde{M}} k_g ds$. The later in turn is related to the boundary conditions that we impose on the surface, i.e. when prescribing a curve at a cutoff-surface near the boundary where the surface $\tilde{M}$ is supposed to be anchored.

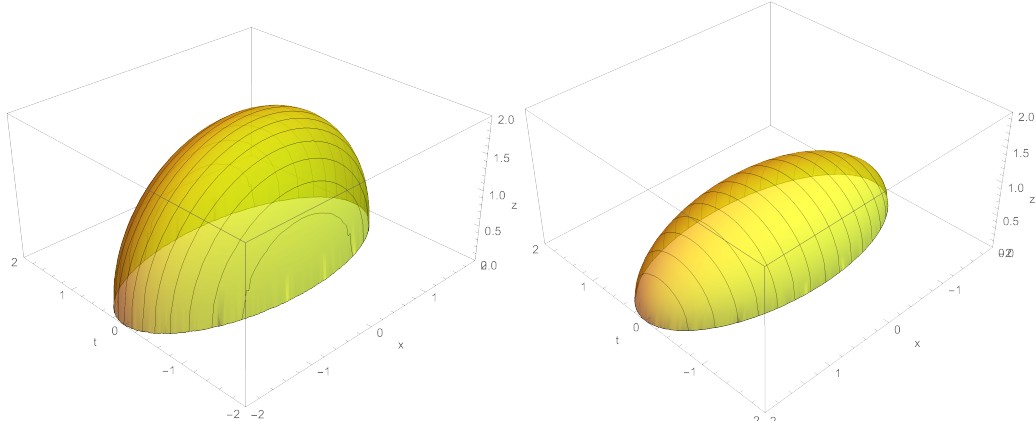

Figure 12: Two hypersurfaces satisfying (71) with the same boundary condition at $z = 0$. For the solution on the left, we find $K < 0$ everywhere, while for the one on the right we find $K > 0$ everywhere.

Let us assume $\chi \geq 0$, which covers the cases of Lorentzian surfaces ($\chi \equiv 0$), disk-shaped Euclidean ones $\chi = 1$ and spherical Euclidean ones $\chi = 2$. As we also assume $R < 0$, that yields a bound

$$V \leq \frac{2\int_{\partial \tilde{M}} k_g \, ds}{-R}. \tag{91}$$

As $k_g$ is the curvature of the edge within the surface $\tilde{M}$, we cannot compute it before having found the surface. However, if that surface is embedded into a larger space with non-vanishing extrinsic curvature, we assume the curvature $k$ of the geodesic within that ambient space to obey $|k| \geq |k_g|$ (a curve in a certain submanifold may be a geodesic with respect to the induced metric ($k_g = 0$) but not the ambient metric ($k \neq 0$)). Equation (91) clearly implies that the average over $k_g$ along the boundary is positive. Assuming now that both $k_g$ and $k$ are positive everywhere, we obtain[12]

$$V \leq \frac{2\int_{\partial \tilde{M}} k \, ds}{-R}. \tag{92}$$

This bound can be computed solely from the boundary conditions, i.e. the curve on a cutoff slice near the asymptotic boundary where we demand the surface $\tilde{M}$ to be anchored. Thus, even though the surfaces we are looking for are not extremal area surfaces, their total volume is bounded from above. Hence, we expect them not to be too "wild" in the bulk, and especially for $\chi \geq 0$ there can be no *smooth* constant negative curvature submanifolds embedded into AdS that don't reach out to the asymptotic boundary. However, in section 5.5 we will study surfaces in AdS that include conical singularities, and they *can* be contained entirely within the bulk.

We will now quickly discuss a potential application of these results, whose full exploitation we however leave to future research. In holography, for example when dealing with the complexity=volume proposal, we are often tasked with finding extremal volume slices in a bulk spacetime. For simplicity, let us consider the case of a Euclidean bulk, where these extremal area slices actually minimise the area. Then, clearly $V_{ext} \leq V$. However, for generic non-translation invariant boundary conditions, the extremal volume slices are not easy to find,

---

[12]This bound can be sharpened again by reinstating the term proportional to $\chi$, assuming $\chi > 0$. Of course, we also have to keep in mind that such constant curvature surfaces may not exist for arbitrary choice of $R$, see e.g. [69].

as seen e.g. in [89] where it was possible to solve the relevant partial differential equation only perturbatively. Hence, our results may be useful in occasions where only a bound on the volume is needed.[13] Not only could one then employ the bound (92) but, as shown in sections 5.2 and 5.3, the constant curvature surfaces can be directly constructed given quite generic boundary conditions (only subject to a symmetry condition) without the need to solve additional differential equations. Some information might then already be gleaned from these surfaces, or they might be used as well motivated initial guess in numerical relaxation schemes.

## 5.5 Lemons in Lorentzian AdS$_3$

In this subsection, we will now put the methods explained in section 5.2 to use in order to construct generic timelike hypersurfaces solving (71) in global Lorentzian AdS$_3$. As we had realised in section 5.4, timelike surfaces embedded into AdS with constant negative curvature can only be fully contained inside the bulk (without boundary) if they have conical singularities, which will of course be the case here. See also appendix B.3 for further details.

It is well known that in Lorentzian global AdS$_3$, there are timelike geodesics that oscillate, i.e. pass through the center of AdS regularly, turning around at finite radial coordinate without ever reaching the boundary. We can now construct co-dimension one hypersurfaces which are foliated by such geodesics, obtaining structures such as the one shown in the top left of figure 13. Specifically, using the global AdS metric (87) (with boundary at $\theta = \pi/2$), the embedding of (the branch valid for $-\pi/2 < t < \pi/2$ of) a radial timelike geodesic is given by

$$t(\theta) = \arctan\left(\frac{E\sin(\theta)}{\sqrt{-1 + E^2\cos(\theta)^2}}\right), \quad \phi = const. \tag{93}$$

where the "energy" $E > 1$ of the geodesic is related to its turning point $\theta_{max}$ by $\theta_{max} = \arccos 1/E$. Following the methods of section 5.2, we can construct surfaces of the form

$$t(\theta, \phi) = \arctan\left(\frac{E(\phi)\sin(\theta)}{\sqrt{-1 + E(\phi)^2\cos(\theta)^2}}\right), \tag{94}$$

where we have promoted $E$ to a $\phi$-dependent parameter. The relevant equations can become a bit cumbersome, but in the special case where $E(\phi) = E = const.$, the induced metric (in $\theta$-$\phi$ coordinates) reads

$$g_{mn} = \begin{pmatrix} -\frac{\sec^2(\theta)}{-1 + E^2\cos^2(\theta)} & 0 \\ 0 & \tan^2(\theta) \end{pmatrix}, \tag{95}$$

while the extrinsic curvature takes the form

$$K_{mn} = \begin{pmatrix} 0 & 0 \\ 0 & E\tan(\theta) \end{pmatrix}, \quad K = E\cot(\theta). \tag{96}$$

Curiously, $K$ diverges at $\theta = 0$ where the surfaces will have a conical singularity.

Due to the presence of these conical singularities at time coordinates $t = 0$ and $t = \pi$ (a consequence of the periodicity of the timelike geodesics), we have adopted the term "lemons" for these shapes.[14] See figure 13 for a number of examples. It is easy to verify that surfaces of the form (94) will automatically satisfy (71), even if $E(\phi)$ is an arbitrary function. Note that $E > 1$, and in the limit $E \to \infty \Leftrightarrow \theta_{max} = \pi/2$, i.e. the surface touches the AdS boundary

---

[13]See [90] for a positivity bound on vacuum subtracted volumes with applications to holography.

[14]Even though similar geometric shapes with a different mathematical definition have been described by the same name [75].

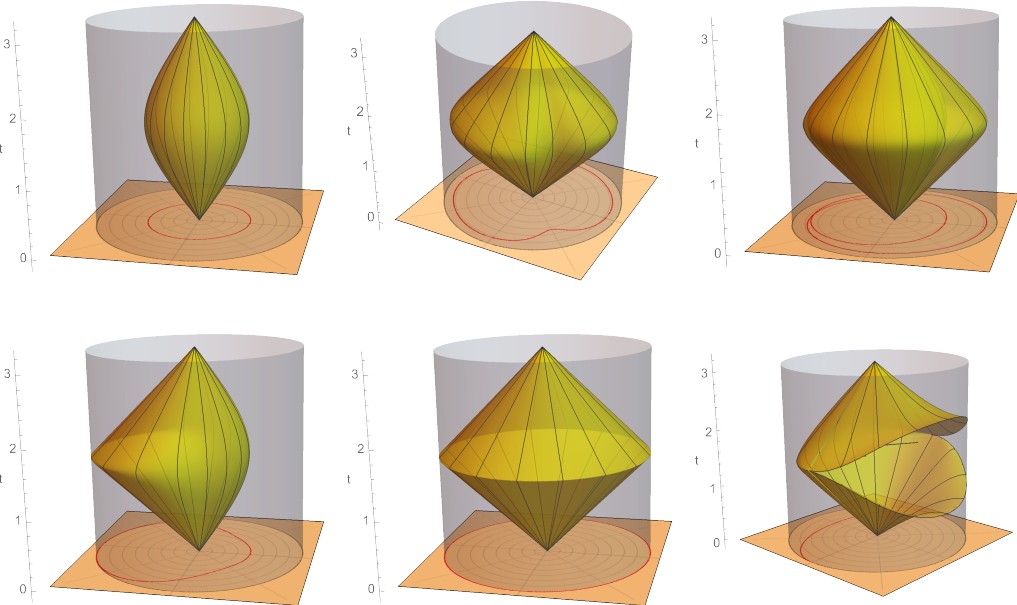

Figure 13: Gallery of generalised lemons. In each plot, we use the coordinate system of (87), where the AdS-boundary is mapped to the grey cylinder at $\theta = \pi/2$. The time axis is shown explicitly. Each yellow surface is an embedding described by (94), with individual timelike geodesics shown as grey lines. In the $t = 0$ plane, the red line indicates the shape of the cut through the surface at its equator. What all of these surfaces have in common is the existence of tips with conical singularities, as demanded by consistency with the Gauss-Bonnet theorem. Apart from this, these surfaces can have not only many different shapes (top left and middle), they can also have self intersections (top right), reach out to touch the boundary (bottom left and middle), or even reach out to intersect the boundary (bottom right). The latter case may actually seem somewhat confusing at first: As discussed earlier in section 5.2, the induced metric has to be maximally symmetric, and hence homogeneous. But evidently, the signature of the metric switches from timelike to spacelike as we travel along the surface, and hence it can not be really homogeneous. This can happen because along the transition line, the induced metric is sufficiently continuous, but not analytic. From top-left to bottom-right, these surfaces are given by $E(\phi) = \sqrt{2}$, $E(\phi) = 2\sin(2\phi) + \cos(4\phi) + 5$, $E(\phi) = 5\sin^2\left(\frac{\phi}{4}\right) + \sqrt{2}$, $E(\phi) = \tan^4\left(\frac{\phi}{2}\right) + \sqrt{2}$, $E \to \infty$, and $E(\phi) = \left(\frac{\cos(\phi)}{\sin(\phi)}\right)^2 + 2$ for $0 < \phi < \pi$, $E(\phi) = -i\left(\left(\frac{\cos(\phi)}{\sin(\phi)}\right)^2 + 2\right)$ for $\pi < \phi < 2\pi$, respectively. The bottom middle figure shows how the WDW patch arises naturally in this construction.

with its equator in this limit. In fact, in this limit $t(\theta, \phi) = \theta$, hence the surface becomes the null boundary (the past part of it for this branch) of the WDW patch of the $t = \pi/2$ time slice on the boundary. This is interesting because the WDW patch, which plays a central role in the complexity=action proposal [20],[15] thus emerges very naturally from our construction. We would like to contrast this with the situation in [28] (see also [29–31] for more recent works in this direction), where the authors introduce a tension term $T$ to their equations as the simplest possible term (however, this tension is then given an a posteriori interpretation as an emergent holographic time). The null-boundaries of the WDW patch for the given boundary time slice are obtained as solution only in the rather unphysical seeming limit $T \to -\infty$. So the fact that the WDW patch arises naturally in our construction is rather encouraging, but as discussed in section 4.4 the null limit for the surface $\tilde{M}$ is related to a divergence in the value of the action. Also, as explained in [92,93], WDW patches in non-translation invariant settings can get quite complicated. So it would be interesting to see whether our method can be adapted to this and help analyse the features of such non-trivial WDW patches by first constructing a foliation of the interiour of the WDW patch in terms of lemon surfaces, and then taking the appropriate limit. Going further, we can even allow imaginary values of $E(\phi)$ in (94) which leads to spacelike surfaces that reach out towards the asymptotic boundary, as also shown in figure 13.

In our calculations motivated by complexity so far, we have always assumed the presence of an initial and final time slices like in figure 9, respectively section 4.4. But as the lemon surfaces start and end on conical singularities, we can as well calculate the action of their interior, without any additional boundary surfaces. There are no joints in this case, hence this only requires the bulk term and the Gibbons-Hawking-York boundary term. We assume the conical singularities to make no contribution to the action, which can be checked by a limiting argument similar to appendix B of [94] where it was shown that caustic points do not contribute to the action.

As (94) describes one half of a lemon, from the conical singularity at $t = 0$ to the equator at $t = \pi/2$ (i.e. from $\theta = 0$ to $\theta = \theta_{max}$ along one branch), the bulk and boundary terms read

$$I_{EH} = \int_0^{\theta_{max}} d\theta \int_{t(\theta,\phi)}^{\pi/2} dt \int_0^{2\pi} d\phi \; \sqrt{-G} \, (\mathcal{R} - 2\Lambda) \,, \tag{97}$$

$$I_{GHY} = 2 \times 2 \int_0^{\theta_{max}} d\theta \int_0^{2\pi} d\phi \; \sqrt{-g} K. \tag{98}$$

where we ignore the common prefactor involving the Newton constant. Together, we find

$$\begin{aligned} &I_{EH} + I_{GHY} \\ &= 8\pi \int_0^{\theta_{max}} d\theta \, \frac{E \sec(\theta)}{\sqrt{E^2 \cos^2(\theta) - 1}} - \frac{\tan(\theta)}{\cos^2(\theta)} \left( \pi - 2 \arctan\left( \frac{E \sin(\theta)}{\sqrt{E^2 \cos^2(\theta) - 1}} \right) \right) \\ &= 4\pi^2 \,. \end{aligned} \tag{99}$$

Hence, for all lemons that do not reach the asymptotic boundary, we obtain the same value for the action. This is not surprising, because in our previous paper [35] we explicitly derived equation (71) as a flow-equation from the bulk-action. The idea was that such a flow might be triggered by turning on a $T\overline{T}$ deformation, moving the boundary into the bulk [38] (see also [95–97]), and the surfaces that satisfy (71) would receive the physical interpretation of being those surfaces on which such a flow can come to rest. As an alternative description,

---

[15]Even more, by causality arguments similar to [91] the WDW patch is actually the largest region in the bulk on which any complexity proposal for one given boundary time slice can depend. For example, the extremal volume slice defining complexity in the complexity=volume proposal is always contained inside of the WDW patch by definition.

these surfaces bound regions of the bulk whose action does not change under infinitesimal deformations of their boundary.[16] But as the interior of the WDW patch can be foliated by such surfaces, and all are valid solutions to the equations of motion, it follows that the action evaluated inside all of these lemons has to have the same value. To provide an analogy, suppose you are looking for the extrema of a potential $V(x)$, where a particle might potentially be at rest, even if unstable. The equation of motion for this is $V'(x) = 0$. If all points $x \in I$ inside an interval satisfy this equation, it follows that the potential is constant in that interval. Concerning the action of the lemon surfaces, this argument is valid not only in the case of constant $E$ as assumed above. It can be checked tediously but explicitly that even for functions[17] $E(\phi)$ the above action calculation yields the same result, as we should now expect.

The solutions of (71) hence have the physical interpretation of defining a foliation of a part of the bulk spacetime in terms of timelike surfaces such that the action inside of each such surface has the same constant value. This also implies that the action evaluated in the region between any two lemons vanishes identically. As said above, based on [35] we hope to interpret these surfaces as potential endpoints of a flow of the asymptotic boundary into the bulk triggered by turning on a $T\overline{T}$ deformation in the boundary theory. Given the time-periodic nature of the lemons, this would clearly have to be done in a time-dependent manner, and it would be interesting to construct such a $T\overline{T}$-deformation explicitly and analyse it from a field theory point of view. Apparently the field theory in question, if it exists, naturally is described by Dirichlet boundary conditions on a bulk submanifold resembling a cyclic universe, starting from an initial (conical) singularity, expanding, contracting, and ending in a final (conical) singularity with a period that has to be exactly $\Delta t = \pi$ before the cycle starts all over again. We leave an investigation of this for future work.

One additional thing that we want to quickly comment on is the action for lemons which do reach the asymptotic boundary of AdS. The WDW patch is obtained in our construction by taking the limit $E \to \infty$ of the lemon surfaces, and as the action inside every lemon for finite $E$ is constant, we might be tempted to assign this value also to the action inside of the entire WDW patch. However, the WDW patch is bounded by null-surfaces which have to be treated in their own special way in action calculations as explained for example in [55], and generally the correct value of the action can not be obtained by a continuous limit from regions bounded by timelike or spacelike surfaces. Nonetheless, some of the terms proposed in [55] for null-boundaries, the so-called counter terms, are not unique (see also [98]) and it has been shown that, for example when translation invariance is slightly broken, they can cause problematic results in the complexity=action proposal [93, 99]. Consequently, there is some interest in alternative methods of treating null-boundaries in the calculation of the bulk action, see e.g. [100]. It might thus be interesting to explore whether there is a well defined alternative prescription for calculating the action inside of WDW patches that would yield the same value that our limiting procedure suggests. Alternatively, one might propose to use timelike lemon surfaces with $\theta_{max} = \pi/2 - \epsilon$, $\epsilon \ll 1$, as a regularisation of the WDW patch and the associated UV divergences that does not need null boundary surfaces, as opposed to using a WDW patch intersected by a cutoff surface at $\theta_{cutoff} = \pi/2 - \epsilon$ which is usually done. Such prescriptions for a modified CA proposal would however yield finite values for complexity, without any $\epsilon$-dependent divergent terms, defying physical expectations for how complexity should behave in a quantum field theory. Turning back to the analysis of the action associated to general lemon surfaces, when $E$ is given an imaginary value, we obtain spacelike surfaces

---

[16]This means that in this paper and [35], it is the Einstein-Hilbert action (including Gibbons-Hawking-York boundary term) itself that acts as the analogue to the functional defined in [71] to describe surfaces of constant curvature in $\mathbb{R}^3$. In fact, that functional was similar to the Einstein action in that it consists of a volume integral (as if $R - 2\Lambda$ was a constant) and an extrinsic curvature boundary term.

[17]We continue to assume, however, that $E(\phi)$ is real and bounded from above, and free from self-intersections, i.e. $E(\phi)$ is a periodic function with period $2\pi$, unlike the third example in figure 13.

that intersect the asymptotic boundary. To calculate the action inside such a "peeled lemon" (like the last example in figure 13) by standard methods, we would have to introduce a cutoff-surface, and the resulting value of the action would be divergent in the limit of vanishing UV regulator $\epsilon$. However, when allowing imaginary values for $E$, we obtain values for the turning radius of the form $\theta_{max} = \pi/2 + i\,\text{arccsch}(Im(E))$. Curiously, this could be taken to suggest that at least on a formal level, the embeddings for the spacelike solutions of (71) can be extended beyond the AdS boundary ($\theta = \pi/2$) by using a complex $\theta$-coordinate, and one might speculate whether such embeddings have an interpretation in terms of "wrong sign" $T\bar{T}$-deformations. For example, by careful analytic continuation the integral (99) can thusly be lifted to a contour integral from $\theta = 0$ to $\theta = \theta_{max}$ in the complex plane, and be shown to still yield the same result even for imaginary $E$. However, it should be pointed out that this complexification approach fails when directly applied to earlier steps in the calculation such as (97) and (98), especially because the volume elements $\sqrt{-G}$, $\sqrt{-g}$ introduce branch-cuts due to the square-roots.

Given the importance of geodesics in our solution method explained in sections 5.2 and 5.5, there appears to be an interesting parallel to the recent work on Lorentzian bit-threads in [101, 102], where likewise geodesic flows were employed. Especially, some of the figures in [102] appear familiar from the construction of lemons in section 5.5. We leave an in depth exploration of possible connections between our work and [101,102] for future research, however for now we want to caution the reader that the apparent connection explained above may only be superficial, for the following reasons: In [102], Lorentzian flows are defined as time-like, divergenceless, future directed vector fields $v$ with a bound on the norm. Such flows are not unique, and using congruences of timelike geodesics is just one convenient way to construct such flows explored in [102], but not the only one. In contrast, in $2 + 1$ bulk dimensions, solutions of (71) have to be foliated by geodesics as we have shown. These geodesics, however, can be both timelike or spacelike. Because of this, there is a difference between how this work and [102] construct foliations of the bulk spacetime outside of the WDW patch, even though the foliations given for the inside of the WDW patch might agree. Furthermore, the similarities end when going to higher dimensions. While the construction of [102] using timelike geodesics still works in higher bulk dimensions, we do not think that such geodesics will have a particular role to play for obtaining solutions of (71) in more than $2 + 1$ bulk dimensions. See also appendix C, where we will study spherically symmetric lemons in higher dimensional global AdS, and quickly comment on their qualitative differences to their lower dimensional counterparts. Of course, the explicit equations given in [101, 102] would still have practical uses in our kind of investigations, e.g. when constructing lemons in the BTZ black hole background.

# 6 Discussion and outlook

In this paper we explored proposals for the cost of path integrals that prepare and transition between states in gravitational theories. We described these path integrals in gravitational theories with Dirichlet boundary conditions on a finite radial surface, which are holographically dual to $T\bar{T}$ deformed CFTs, and gave the precise map between path integrals in the bulk and the boundary. We have given bulk proposals for the cost of such path integrals that satisfy a set of physical requirements, and shown explicitly how such path integrals can be optimised: by minimising their cost over a suitable set of bulk subregions to reduce to existing holographic state complexity proposals. Lastly we developed general methods for gravitational action-type proposals.

Our work was partly inspired by the idea that holographic complexity proposals and their

possible generalizations originate from coarse-graining circuits represented by moving the asymptotic boundary inwards [103]. See figure 1. Such an approach to holographic complexity of bringing in the asymptotic AdS boundary to finite cutoff can be made precise through the language of $T\overline{T}$-deformed holographic CFTs [35,38,53,62]. The aim of our work was to generalise this approach to consider general bulk subregions within finite cutoffs, functions on which we propose to be the cost of the path integral on the subregion.

Our approach is complementary to and generalises existing work on holographic state complexity. Maximum volume slices and WdW patches from our perspective are bulk subregions that minimise path integral cost for a suitably chosen proposal. Holographic complexity arises from the optimisation of path integral preparation of states. Note that once the function on bulk subregions is fixed, the 'optimal' subregion that minimises the path integral cost is dynamically determined; we do not independently specify the optimal bulk subregion *and* the function on it. This is in contrast to the two-functional holographic complexity proposals pursued in [22], and it would be worthwhile to combine their approach, complexity=anything, with ours, cost=anything, and see what subset of their proposals arise from the minimisation of carefully chosen path integral cost proposals over suitable bulk subregions.

In section 4 we were able to find path integral cost proposals that reduce to some of the existing holographic state complexity proposals. Cost = boundary volume in a Euclidean bulk reduces to complexity = volume at the time reflection symmetric slice. We gave an physical justification for this proposal in terms of a $T\overline{T}$-motivated notion of discretisation of the boundary path integral. Cost = bulk volume in Lorentzian signature reduced to complexity = volume 2.0. We also showed that, in the special case of pure global Eucldiean AdS, the cost=gravitational action proposal from our previous paper [35] reduces to complexity=volume, though again only on a slice that is time reflection symmetric. Lastly, and much in the spirit of [22], we applied our cost=anything philosophy to conjecture novel complexity proposals. Our new codimension-0 candidate holographic complexity proposal satisfies at least persistent linear growth in thermofield double states, though the proposal was not derived by minimising a cost, but rather through providing new covariantly defined boundary anchored bulk regions.

We were not able to find a path integral cost proposal that reduces to the complexity=action conjecture, or the complexity=volume conjecture except on time symmetric slice where we are free to analytically continue between Euclidean and Lorentzian signature. The key issue is the existence of Lorentzian bulk subregions for which the gravitational action is unbounded in both directions, which prevents the use of Lorentzian gravitational action as a cost proposal that reduces to complexity=action. Our analysis does not rule out the complexity=action proposal, since failure to find a suitable cost proposal does not prove its non-existence. Furthermore, the shortest path in the Hilbert space may involve non-geometric states, which would be outside the scope of our work.

In section 5 we turned our attention to the equations of motion specifying boundary surfaces extremizing gravitational action in our cost = (Euclidean) action proposal [35], see equation (71). While in our previous paper we discussed easily-obtainable homogeneous solutions of these equations, here we were able to solve these equations in generality in AdS$_3$ geometries using that such surfaces are generated (foliated) by bulk geodesics. Our observation applies both to the Euclidean and the Lorentzian bulk spacetimes. We also showed that the boundaries extremizing gravitational action provide an interesting new way of foliating Wheeler-DeWitt patches that we call 'lemons', see figure 13.

There a several avenues for future research. Our discussion of bulk cost functions has been entirely phenomenological. In the enormous set of cost proposals that satisfy our physical requirements, we have given no reason to favour one over another, besides perhaps simplicity. Moreover, except for cost=boundary volume we have given no physical justification for any of our proposals. That said, cost and complexity are inherently ambiguously defined, so even

if one could find a gate set and metric on the space of operators that gives one of our cost proposals, that would not favour that proposal over others as there is no reason to favour that definition of cost over others. We view the size of our set of cost proposals as directly related to the inherent ambiguity of definition. As further justification, note that the set of complexity proposals that satisfy reasonable requirements is similarly enormous [22].

In our work we predominantly considered bulk subregions of single, fixed bulk on-shell geometries in the semiclassical limit. When considering semiclassical path integral preparations of and transitions between bulk states one should in principle consider all on-shell geometries that satisfy the boundary conditions. We did not do so for technical ease, and we were justified in doing so by carefully choosing our cost proposals such that there was no benefit in altering the geometry in the interior of the cost-minimising bulk subregion, or it was not possible without violating the boundary conditions. It would be interesting to allow for different bulk geometries as well as different subregions thereof in our path integral optimisation. Similar perspectives were pursued in [104], and having a good control of the bulk dual to a circuit might allow to gain a microscopic understanding into the meaning of at least some bulk cost proposals.

We have considered cost proposals for Lorentzian and Euclidean bulks, but we could also consider more general complex metrics. Such metrics are relevant, for example, in calculating the temperature of Kerr black holes using path integral methods [105]. See also [106] for a recent discussion of complex saddles in gravity. For a more complete story we should have cost proposals for such path integrals. Cost should be real-valued, which limits the possibilities of functions on subregions of complex manifolds. When the complex manifold is related to a real one by Wick rotation of a stationary spacetime then the volume form remains real, so volume may be a reasonable cost proposal in certain bulks with complex metrics.

We required our cost proposals to be covariantly defined, which includes independence from choice of time foliation, see section 3.2 and figure 5. One could in principle relax this and allow for cost proposals that depend on a time-foliating vector field as was done in [32].

Another question is related to a potential intrinsic difference between Lorentzian and Euclidean circuits. In the Euclidean case due to exponential suppression certain contributions to path integrals become very small and, in practice, negligible. As a result, part of the optimization process might be related to eliminating such practically negligible contribution, see [107] for a free QFT realisation. In the Lorentzian case, the problem of optimization becomes an algebraic problem of exact decomposition of operators into a sequence of gates (circuits). It would be very interesting to understand if one should view the Euclidean problem also as an exact problem, and so lose all the benefits Euclidean time evolution brings for state preparation. This is the perspective pursued by [32], or to point out where in the gravity description this difference between the Euclidean and Lorentzian case manifests itself.

## Acknowledgments

We are grateful to Antony Speranza for useful discussions. The work of MF is supported through the grants CEX2020-001007-S and PGC2018-095976-B-C21, funded by MCIN/AEI/10.13039/501100011033 and by ERDF A way of making Europe. SH acknowledges support from the Ramon Areces Foundation (Spain) and the ERC Consolidator Grant CoG 772295 "Qosmology". ARC and JdB are supported by the European Research Council under the European Unions Seventh Framework Programme (FP7/2007-2013), ERC Grant agreement ADG 834878. AR is supported by NWO-I through the Scanning New Horizons (16SNH02) program.

# A  Cost equals gravitational action in Lorentzian Poincaré AdS$_3$

Let us calculate the gravitational action of the subregion of Lorentzian Poincaré AdS$_3$ depicted in figure 9. The gravitational action is

$$I = I_{EH} + I_{GHY} + I_{Hayward}\,. \tag{A.1}$$

The Einstein-Hilbert term is

$$I_{EH} = \frac{1}{16\pi G_N} \int \sqrt{-G}(\mathcal{R} - 2\Lambda)\,, \tag{A.2}$$

where $\mathcal{R} = -6$ and $\Lambda = -1$ in AdS$_3$ with $L = 1$. Then $I_{EH}$ is proportional to the spacetime volume,

$$I_{EH} = -\frac{1}{8\pi G_N} \int dx \int_{t_i}^{t_f} dt\, \frac{1}{\rho^2}\,. \tag{A.3}$$

Our region has two corners, both of which are spacelike surfaces meeting a timelike surface, both of which contribute to the gravitational action[18]

$$I_{Hayward} = \frac{1}{8\pi G_N} \int \sqrt{\sigma}\,\eta\,, \tag{A.4}$$

where $\sigma$ is the induced metric on the joint, and

$$\sinh\eta = -t_1 \cdot n_2\,, \tag{A.5}$$

where $t_1$ is the (timelike) normal to the $t = \{t_i, t_f\}$ slices, and $n_2$ is the (spacelike) normal to $z = \rho(t)$. For our setup it's easy to show that

$$\eta = \pm \operatorname{arctanh}\dot\rho\,, \tag{A.6}$$

with $+(-)$ at the $t_f (t_i)$ joint. This gives

$$I_{Hayward} = \frac{1}{8\pi G_N} \int dx \left[ \frac{\operatorname{arctanh}\dot\rho(t_f)}{\rho(t_f)} - (t_f \leftrightarrow t_i) \right]\,. \tag{A.7}$$

Let us calculate the contribution of the finite cutoff time-like boundary to the gravitational action through the Gibbons-Hawking-York (GHY) term. The boundary is the hypersurface

$$z = \rho(t)\,, \tag{A.8}$$

in Poincaré AdS$_3$ (55). We want to calculate the extrinsic curvature of this surface

$$K = \nabla_\mu n^\mu = z^3 \partial_\mu (z^{-3} n^\mu)\,, \tag{A.9}$$

where the unit normal to the hypersurface is given by

$$n^\mu := \frac{G^{\mu\nu}\zeta_\nu}{|\zeta|}\,, \tag{A.10}$$

with un-normalised normal

$$\zeta_\mu = \partial_\mu(\rho(t) - z);\quad \zeta_x = 0\,,\quad \zeta_t = \dot\rho\,,\quad \zeta_z = -1\,, \tag{A.11}$$

---

[18]See appendix A of [66] for Hayward corner terms for every kind of corner.

and

$$|\zeta| = z\sqrt{1-\dot{\rho}^2}\,. \tag{A.12}$$

Plugging these in to the formula for $K$ gives

$$K|_{z=\rho} = \frac{-\rho\ddot{\rho} + 2(1-\dot{\rho}^2)}{(1-\dot{\rho}^2)^{3/2}}\,, \tag{A.13}$$

which is exactly what you get if you Wick rotate the Euclidean answer, so that $\dot{\rho}^2 \to -\dot{\rho}^2$, and $\ddot{\rho} \to -\ddot{\rho}$. The GHY term is

$$
\begin{aligned}
I_{GHY} &= \frac{1}{8\pi G_N} \int \sqrt{-g}K \\
&= \frac{1}{8\pi G_N} \int dx \int_{t_i}^{t_f} dt \frac{-\rho\ddot{\rho} + 2(1-\dot{\rho}^2)}{\rho^2(1-\dot{\rho}^2)}\,.
\end{aligned}
\tag{A.14}
$$

Integrating the double derivative term by parts gives a boundary contribution that cancels the Hayward terms in our case (however, this cancellation would not have happened with general spacelike boundaries rather than the constant time slices we have considered). Combining everything gives

$$I = \frac{1}{8\pi G_N} \int dx \int_{t_i}^{t_f} dt \left(\frac{1 - \dot{\rho}\,\text{arctanh}\,\dot{\rho}}{\rho^2}\right)\,, \tag{A.15}$$

which agrees with the Wick rotation of the Euclidean result of [35]. We note that while the GHY term (A.14) by itself may remain finite when a null-limit of the surface $\rho(t)$ is taken (see [108]), the full action (A.15) diverges in this limit. This can be shown to be a consequence of the Hayward-type corner terms (A.4). In general, the procedure of obtaining gravitational action of a region with null boundaries as a null-limit of timelike or spacelike regions is ambiguous, see [55].

## B  Conical singularities in the Gauss-Bonnet formula

### B.1  Euclidean case

While most sources don't state the Gauss-Bonnet theorem explicitly including terms necessary for conical singularities, it is not hard to find the appropriate terms. While a formal proof was given in [84], a less rigorous but quite simple approach would be the one of [109] where it was simply postulated that the Gauss-Bonnet theorem should continue to hold in the presence of conical singularities, and then the necessary correction term was derived by looking at one simple example.[19] In this section, we will give our own argument which easily generalises to the Lorentzian case in the next subsection.

To do so, we consider a body with a conical singularity like the one sketched on the left side of figure 14. Of course we know that a conical singularity can be resolved, in a sense, by introducing a cut and spreading the cone on a flat plane as indicated in the figure. Hence, let us now assume that, as indicated by the dashed red lines in the figure, we introduce a cut that goes from the exact location of the conical singularity to a point elsewhere in the surface

---

[19]This can be justified by observing that at least for the symmetric conical singularities on surfaces of revolution, all conical singularities are locally equivalent up to their deficit angle, and hence the correction term should only depend on the deficit angle.

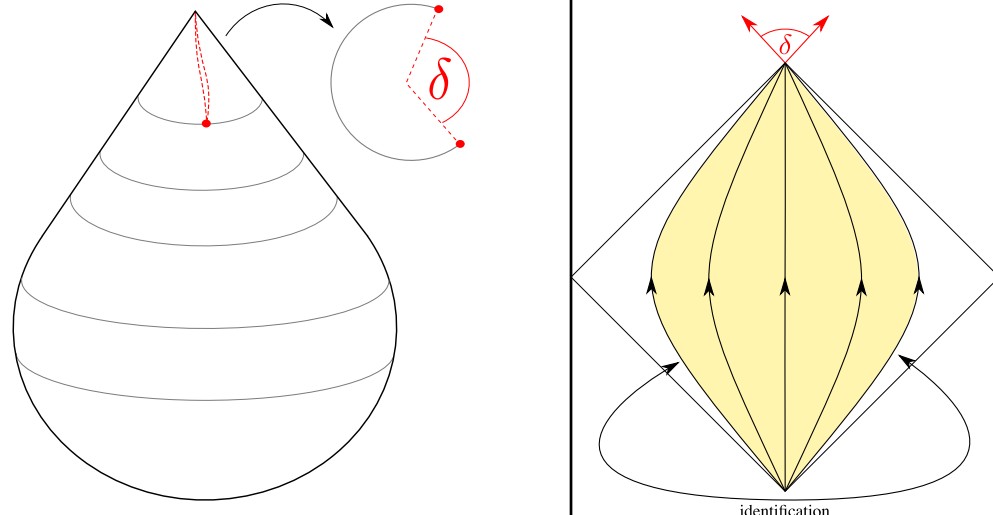

Figure 14: Left: Illustration for the derivation of the Gauss-Bonnet-theorem in the presence of conical singularities. We consider a two dimensional surface, either Euclidean or Lorentzian, with a conical singularity, and introduce a cut (red dashed lines) from the conical singularity to a point elsewhere in the surface, where it is locally smooth. The conical deficit of the singularity (or its Lorentzian analogue) is $\delta$. Right: Construction of the lemon's induced metric by taking a region between two timelike geodesics in AdS$_2$ and identifying the boundary. At the intersection of the two geodesics that form the boundary of the region, their two tangent vectors cross with a relative boost factor $\delta$, which is the Lorentzian analogue of the deficit angle for the conical singularity that is formed at this location due to the identification. The two vertical lines indicate the AdS$_2$ boundaries at $\theta_2 = \pm\frac{\pi}{2}$.

where it is locally smooth. Before introducing this cut, we assume the Gauss-Bonnet theorem holds in a form

$$\int_{\tilde{M}} \frac{R}{2} dV + \int_{\partial \tilde{M}} k_g ds + \sum_{\text{(old) corners } c} \alpha_c + X_{\text{conical sing.}} = 2\pi\chi \,. \tag{B.1}$$

Herein, $X_{\text{conical sing.}}$ is the as of now unkown contribution from the conical singularity which we want to derive.

How does introducing the cut change both sides of this equation? The first term stays the same because we don't assume to spread the cut open by deforming the surface, both edges of the cut remain at the same location. We have merely indicated a slight spread of the cut as a visual aid in the figure. Due to the cut, new contributions to the second term could in principle appear, however we argue this won't matter for multiple reasons. Firstly, the contributions from the two sides of the cut should cancel exactly. Secondly, we could choose the cut to be geodesic, setting $k_g = 0$. Thirdly, we can take a limit where the cut is infinitesimally short. The fourth term will not be present anymore due to the resolution of the conical singularity, hence on the left hand side all changes come down to the additional terms due to the two corners at both ends of the cut. Of course, introducing the cut also causes a change of the topology like removing a disk, and due to the behaviour of the Euler characteristic under connected sums

this means $\chi$ is reduced by 1. Hence, we find

$$\int_{\tilde{M}} \frac{R}{2} dV + \int_{\partial \tilde{M}} k_g \, ds + \sum_{\text{(old) corners } c} \alpha_c + \sum_{\text{new corners } c} \alpha_c = 2\pi(\chi - 1). \tag{B.2}$$

Comparing (B.1) and (B.2), we find that the correct contribution for conical singularities is hence determined by the contributions for corners along boundaries via

$$X_{\text{conical sing.}} = \sum_{\text{new corners } c} \alpha_c + 2\pi. \tag{B.3}$$

So what are now the contributions from the two corners at which the cut starts and ends? Firstly, at the point in a locally smooth neighbourhood of the surface, essentially the new boundary introduced by the cut makes a 180-degree turn there, i.e. $\alpha_{c1} = -\pi$. Note the negative sign because this corner is a concave one from the point of view of the surface. The corner located at the position of the conical singularity is also concave from the point of view of the surface, but there, with respect to the local geometry, the angle by which the boundary changes its direction is reduced by the deficit angle $\delta$ of the conical singularity as evident from the figure 14. Hence, the contribution is $\alpha_{c2} = -(\pi - \delta)$, and we find

$$X_{\text{conical sing.}} = \alpha_{c1} + \alpha_{c2} + 2\pi = \delta. \tag{B.4}$$

This means that the contribution of a conical singularity in the Gauss-Bonnet theorem should be simply its deficit angle $\delta$, as also realised in [84, 109]. Our derivation makes it obvious why the terms coming from conical singularities and terms coming from corners of the boundary are so similar (just sums over angles), and can readily be generalised to the Lorentzian case as we show in the next subsection.

## B.2 Lorentzian case

Generalisations of the Euclidean Gauss-Bonnet theorem to the Lorentzian case were worked out in [85–88],[20] and while none of these papers explicitly discusses conical singularities, the generalisation of the concept of an angle to the Lorentzian case lies at the heart of all of these works. As the appropriate terms for conical singularities are just sums over deficit-angles which can be derived from the terms needed for boundaries with corners, as shown in the previous subsection, it is hence easy to generalise this also to the Lorentzian case. The papers [85–88] differ in some of the details of exactly how to define Lorentzian angles, e.g. some use complex quantities, so for concreteness we follow [85] and define the (always real valued) oriented Lorentzian angle or boost parameter $\delta$ between two future pointing normalised timelike vectors $X$ and $Y$ to satisfy[21]

$$\cosh(\delta) = -X \cdot Y. \tag{B.5}$$

The Lorentzian Gauss-Bonnet-theorem then takes the form [85]

$$\int_{\tilde{M}} \frac{R}{2} dV + \int_{\partial \tilde{M}} k_g \, ds + \sum_{\text{corners } c} \alpha_c = 0, \tag{B.6}$$

where some care has to be taken concerning the signs of the generalised angles $\alpha_c$. In fact, the difference to the Euclidean case is two-fold: Firstly, the right hand side automatically vanishes

---

[20]See also [110] for an analysis of the Gauss-Bonnet theorem for surfaces of varying signature.

[21]For our specific case of future pointing timelike vectors, this follows from the more general equations given in [85] by using the relation $\cosh(\log(x)) = \frac{1+x^2}{2x}$, the normalisation of the vecors, and some algebra.

($\chi \equiv 0$), secondly, traversing a closed timelike geodesic polygon in flat space yields the total Lorentzian angle

$$\alpha_{12} + \alpha_{23} + ... + \alpha_{n1} = 0 \,, \tag{B.7}$$

whereas in the Euclidean case the exteriour angles of a polygon sum to $2\pi$. This means that we can quite easily generalise our derivation from the previous subsection to the Lorentzian case, however while there on both the left- and the right-hand side an additional term $2\pi$ appeared, this will not be the case in the Lorentzian setting, and we find that the appropriate contribution to (B.6) to account for Lorentzian conical singularities will be a term $X_{\text{conical sing.}} = \delta$ where $\delta$ is the Lorentzian analogue of the deficit angle at the conical singularity.

## B.3 Application to Lemons

Let us now demonstrate how the Lorentzian version of the Gauss-Bonnet theorem (including terms for conical singularities) can be applied to the example of a lemon surface from section 5.5, where for simplicity we will assume a $\phi$-independent parameter $E$. To do this, we view the lemon as a boundary-less closed surface which however has two conical singularities, as e.g. the example on the top-left of figure 13. Hence as $\chi \equiv 0$ in the Lorentzian case, we need to verify

$$\frac{RV}{2} + \delta_{\text{past conical sing.}} + \delta_{\text{future conical sing.}} = 0 \,. \tag{B.8}$$

To correctly calculate the Lorentzian analogue of the deficit angle, the easiest way in this case (but not necessarily the only or most general one) is to resolve the conical singularity by introducing a cut. For this, consult the right side of figure 14. As we showed in section 5.2, the induced metric on the surface should be locally AdS$_2$. But of course, when thinking about AdS space we usually envision a static spacetime with an asymptotic boundary as opposed to something resembling a periodic cosmology that starts from an initial (conical) singularity, expands, contracts, and ends in a final (conical) singularity, like the surfaces shown in figure 13. The resolution of this issue is of course that the induced metric of the lemons is only locally AdS, and we know that global identifications can yield very non-trivial geometries, like for example the BTZ black hole. The right side of figure 14 shows how an identification applied to (global) AdS$_2$ with line element

$$ds^2 = \frac{1}{\cos(\theta_2)^2} \left( -dt_2^2 + d\theta_2^2 \right) \,, \tag{B.9}$$

can yield the induced geometry of a lemon surface. Note that we have introduced coordinates $t_2, \theta_2$ on AdS$_2$ to distinguish them from the coordinates of global AdS$_3$ (87), which in this section we explicitly replace by $t \to t_3$, $\theta \to \theta_3$. Also note that (87) and (B.9) have the same AdS-radius (which we set to one), hence the three dimensional Ricci scalar is $\mathcal{R} = -6$ and the two dimensional one is $R = -2$, as required by our construction (e.g. (72)).

To create the lemon, we have to take the region between two intersecting timelike geodesics in AdS$_2$ and then identify these two geodesics. For concreteness, we assume both boundary geodesics in figure 14 to turn around at a maximal radial coordinate $|\theta_{max,2}| = \arccos(1/E_2)$ according to the coordinate system (B.9). Concerning the lemon surface embedded into the AdS$_3$ ambient space with coordinate system (87) as shown in figure 13, we introduce the turnaround radius $\theta_{max,3} = \arccos(1/E_3)$. These two sets of parameters are related because the *diameter* of the AdS$_2$ region (the shaded region in figure 14) has

to be equal to the *circumference* of the surface when embedded into AdS$_3$ (the surfaces in 13). This yields the relation

$$4\mathrm{arctanh}\left(\tan\left(\frac{\theta_{max,2}}{2}\right)\right) = 2\pi \tan(\theta_{max,3}). \tag{B.10}$$

The first term in (B.8 ) is easy to compute from the induced metric (95), and we find

$$\frac{RV}{2} = -4\pi\sqrt{E_3^2 - 1} = -4\pi\tan(\theta_{max,3}). \tag{B.11}$$

For the evaluation of $\delta_{\text{future conical sing.}}$ (which equals $\delta_{\text{past conical sing.}}$ by symmetry), we note that at the point ($\theta_2 = 0$) where the two boundary geodesics intersect, their future pointing normalised tangent vectors (drawn read in figure 14) read (this can be shown from (93))

$$X_{\pm}^m = \left(\begin{array}{c} X_{\pm}^{t_2} \\ X_{\pm}^{\theta_2} \end{array}\right) = \left(\begin{array}{c} E_2 \\ \pm\sqrt{E_2^2 - 1} \end{array}\right), \tag{B.12}$$

and hence are boosted with respect to each other by a boost parameter/Lorentzian angle

$$\delta_{\text{future conical sing.}} = \mathrm{arccosh}\left(-X_+ \cdot X_-\right) = \mathrm{arccosh}\left(2E_2^2 - 1\right) = 2\pi\tan(\theta_{max,3}), \tag{B.13}$$

even though by the identification of the two boundary geodesics also these two vectors are formally identified. With (B.11 ) and (B.13 ), we verify that (B.8 ) is satisfied as required.

## C  Lemons in higher dimensions

Let us try to find analogues of the lemon surfaces studied in section 5.5 in global Lorentzian AdS$_4$,

$$ds^2 = \frac{1}{\cos(\theta)^2}\left(-dt^2 + d\theta^2 + \sin(\theta)^2 d\psi^2 + \sin(\theta)^2\sin(\psi)^2 d\phi^2\right), \tag{C.1}$$

with boundary at $\theta = \pi/2$. Assuming rotational symmetry, we just have to propose an embedding parametrized as

$$t(\theta, \psi, \phi) = f(\theta). \tag{C.2}$$

After some computations, equation (71) then yields the ODE

$$4\cot(\theta)f'(\theta)f''(\theta) - 2\left(\csc^2(\theta) + 2\right)f'(\theta)^2\left(f'(\theta)^2 - 1\right) = 0, \tag{C.3}$$

which effectively is a *first order* ODE for $f'$, as $f$ does not show up in the equation. We find the solution:

$$f'(\theta) = \frac{\sqrt{\sin(2\theta)}\cot(\theta)}{\sqrt{C + \sin(2\theta)\cot^2(\theta)}}. \tag{C.4}$$

Unfortunately this is hard to integrate to get an analytic expression for $f$. Nevertheless, we can distinguish three cases, see also figure 15. For $C > 0$, $f' \leq 1$ and the curve $f(\theta)$ we obtain is spacelike and reaches all the way to the boundary at $\theta = \pi/2$. For $C < 0$, $f' \geq 1$ and the curve $t = f(\theta)$ we obtain is timelike and $f'$ diverges at some finite $\theta_{max} \leq \pi/2$, the turning point of the surface. For $C = 0$ we get $f' = 1$, i.e. we obtain the null boundary of the WDW patch in this limit. This means that these spherically symmetric lemons in AdS$_4$ (and also

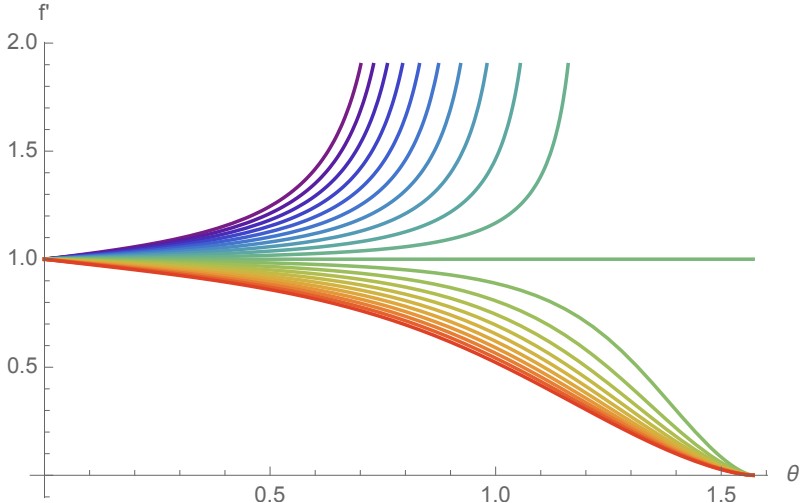

Figure 15: Equation (C.4) for $C$ varying in equal steps between $-1$ (blue) and 1 (red). As before, the AdS$_4$ boundary is at $\theta = \pi/2$.

higher dimensions as we have verified) will share many qualitative features with their AdS$_3$ counterparts, but there are also some interesting qualitative differences that make the AdS$_3$ lemons special.

First of all, while ansatz (73) would of course also work in higher dimensions, it is not generic in these cases, and in fact the extrinsic curvature tensor of the solutions discussed here will not have this form. Consequently, the higher dimensional lemons are not foliated by timelike geodesics of the ambient AdS space. Furthermore, note that $f'(0) = 1$ for any $C$ in (C.4), so at the center the embeddings will always approach the local lightcone. We have discussed the appearance of a conical singularity already in section 5.5, but there for finite turning point radius the embedding at the conical singularity did not approach the lightcone. What this means is that unlike the AdS$_3$ case, for AdS$_{d \geq 4}$ the metric will degenerate close to the conical singularity and this is accompanied by the appearance of a *curvature singularity* of the induced metric there. While $R$ is of course constant by construction, this happens for example for the Kretschmann scalar. Another qualitative difference between AdS$_3$ lemons and the higher dimensional case is that the former all neatly fit into a time interval of size $\Delta t = \pi$ as shown in figure 13. This is because the periodicity of the timelike AdS geodesics is independent of the parameter $E$ introduced in section 5.5. In contrast, integrating (C.4) numerically shows that the AdS$_{d \geq 4}$ lemons will have different sizes, depending on how close to the boundary they reach.

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
