# Peer review of "Cost of holographic path integrals"

_SciPost Physics, doi:SciPost Phys. 14, 061 (2023)_

## Round 2 · Referee Report · Anonymous (Referee 1) · 2022-8-3

Strengths

1-Present general holographic proposals for the cost of path integrals
2-Potential for more applications/studies in this direction

Weaknesses

1-The physical interpretation of some cost proposals is not clear. 2-It is hard to get the connections of section 5 to previous sections. 3- Some arguments/conclusions are confusing. 4- It is better for most readers if the author can focus on one specific cost function and study it in more detail.

Report

This paper explores proposals for the cost of holographic path integrals. The gravitational region with a finite cut-off surface describes the holographic path integrals representing the transition amplitude between states on two Cauchy slices. Moreover, the authors also show that some cost functions associated with specific bulk subregions can reduce to holographic complexity proposals, e.g., CV, CV2.0, CA. In the last part, there are more detailed studies about the surfaces with constant intrinsic curvature.

This submission provides new proposals and also new results. It meets the acceptance criteria of SciPost Physics. But I have to say I have a hard time understanding some content of this paper. I feel that the conclusions or even the cost proposals are not well illustrated. There are two many different/independent examples. Most of them are briefly discussed. It would be better for readers like me if the authors could discuss one cost function in more detail or compare different cost functions in one specific bulk spacetime (like AdS$_3$ in either Poincare coordinates or global coordinates). This is just a suggestion rather than a request.

In the following, I list some questions which should be answered or explicitly illustrated in the paper.

  1. In Table 1, the authors summarize the properties of the path integral cost proposals, which contain both Euclidean spacetime and Lorentzian spacetime. What is the meaning of this Lorentzian bulk spacetime from the viewpoint of bulk path integrals? What are the differences between the Euclidean and Lorentzian signatures?

  2. The authors stress that the path integral cost is expected to be UV finite in many places of this paper. As shown in Figure 1, this is obvious from bulk spacetime since there is a finite radial cut-off surface associated with Dirichlet boundary conditions. However, it is described in section 4 that the cost proposal could reduce to holographic complexity proposals, which are UV divergent. Does this reduction still get involved with a finite cut-off surface and result in a finite holographic complexity of a state?

  3. In section 2.2 for bulk path integrals: the bulk subregion is given by the interior enclosed by two time slices $\Sigma_1, \Sigma_2$ and a cut-off surface $\tilde{M}(\lambda)$. The optimization for the bulk path integrals is performed by changing the shape of $\tilde{M}(\lambda)$ but fixing the condition in equation (2.15). In this optimization process, I believe that the two slices $\Sigma_1, \Sigma_2$ are fixed because it is related to the initial/final state. Is the codimension-two surfaces $\partial \Sigma_1, \partial \Sigma_2$ are always fixed (at a finite cut-off radius)? What happens if we push $\partial \Sigma_1, \partial \Sigma_2$ to the asymptotic boundary? This is also related to my confusion about the UV divergence of the cost function which is reduced to holographic complexity.

  4. The authors state that cost proposals could involve any codimension-n subregions. I can understand that the cost is defined on a bulk codimension-zero subregion, a codimension-one surface like $\Sigma_i$ or codimension-two regions like $\partial \Sigma$. But why other lower dimensional regions can play a role in the bulk path integrals?

  5. In order to interpret the cost proposal as the complexity of a state, an important step is taking $\Sigma_1$ to a point with zero volume. From my understanding, this point is referred to as the middle of AdS bulk (see Figure 2). Is the choice of position of this point $\Sigma_1$ arbitrary (because of isometries of AdS or asymptotically AdS)? It is argued below equation (4.12) that one can get half of the WDW patch by shrinking one state to a bulk point. However, I believe that the position of this point with respect to the boundary of $\Sigma_2$, i.e., $\partial \Sigma_2$ is important to recover half of the WDW patch. My concern for this case (in section 4.2) is that the WDW patch is fixed by a given slice $\Sigma_2$. In other words, the tip of the WDW patch is completely determined by $\Sigma_2$. So from my viewpoint, one has to carefully choose the bulk point $\Sigma_1$ to get a WDW patch. But this sounds unreasonable. Moreover, let us consider AdS black hole background. The WDW patch at late times is not a square because the null boundaries hit the singularity at the interior of the black hole. How to illustrate this situation by using the cost proposal?

  6. In section 4.1, the authors argue that the cost proposal in equation (4.2) reduces to CV because $\tilde{M}^\ast$ is an extremal hypersurface with $K=0$. I get confused by this statement. I thought that $\tilde{M}$ (also $\tilde{M}^\ast$) is a kind of finite cut-off surface and it is {\it timelike} in Lorentzian signature, . However, the content around equation (4.4) says that $\tilde{M}^\ast$ is a spacelike time slice with $K=0$.

  7. The equation (4.24) is derived from equation (4.23) by taking $T=0$. What is the meaning of this limit? I thought that the value $\tau_2 = T$ is considered as an initial condition which is fixed.

  8. It is better to add more explanation at the beginning of section 5. I feel that that section is not firmly connected to the content in previous sections.

  • validity: good
  • significance: good
  • originality: high
  • clarity: ok
  • formatting: good
  • grammar: excellent

Author:  A. Ramesh Chandra  on 2022-10-06  [id 2889]

(in reply to Report 1 on 2022-08-03)
Category:
answer to question
correction

We thank the referee for the very useful comments and questions. We address them below, numbered in accordance with the referee's questions.

  1. The key similarities and differences between the Euclidean and Lorentzian bulk path integrals are explained in section 2.2. in the sentence starting ``We consider bulk path integrals in both Lorentzian or Euclidean signature ...". Lorentzian path integrals prepare unitary operators while Euclidean path integrals create (un-normalised) density operators. We consider cost proposals in Euclidean and Lorentzian signature separately because the operators which the path integrals prepare are fundamentally different.

  2. The referee is correct that with a finite cut-off surface the holographic complexity proposals one finds by reducing our holographic cost proposals are UV-finite. This is not a problem: a UV-finite state complexity is not unreasonable in a $T\bar T$ deformed CFT. A UV-divergent state complexity can be found by taking the cutoff surface towards the asymptotic boundary which corresponds to turning off the $T\bar T$ deformation.

  3. For fixed intial and final states, the codimension-2 surfaces $\partial\Sigma_1$ and $\partial\Sigma_2$ are fixed. By pushing $\partial\Sigma_1$ and/or $\partial\Sigma_2$ to the asymptotic boundary we remove the cut-off on the inital and final states. This would also result in a UV-divergent cost and complexity.

  4. We agree that dimensions lower than codimension-2 might be unnatural from the viewpoint of bulk path integrals. Since our approach was to consider all proposals for a bulk cost, we allowed for regions of all codimensions.

  5. It was incorrect to call $\Sigma_1$ the empty set, as we do for example in section 2.3, and we will change where we have written empty set. As stated earlier in the paper, the initial state is defined on $\Sigma_1$, so we need a non-empty $\Sigma_1$ to define a non-trivial initial state. When relating cost to state complexity, on the field theory side, we take the initial state to be a highly coarse-grained state, in the Hilbert space of a $T\bar T$-deformed CFT with a large deformation parameter. In the bulk, this corresponds to a $\Sigma_1$ which is small - a size parametrically smaller than $\Sigma_2$, the surface on which the less-coarse-grained state we are preparing lives. The referee is concerned that the position of $\Sigma_1$ with respect to $\Sigma_2$ must be fine-tuned, in particular at the tip of the WDW patch. It is true that the WDW patch is fixed by $\Sigma_2$, and that in order to \textit{exactly} land on WDW patch fine-tuning would be needed - our counterargument is that, with a scale separation in size between $\Sigma_2$ and $\Sigma_1$, the minimised cost is insensitive at leading order in the UV cutoff to the size and position of $\Sigma_1$. As an example of the insensitivity, at leading order, of the minimal cost to the relative size and position of $\Sigma_1$, consider the cost equals Lorentzian codim-0 bulk volume proposal, which we claimed minimises (over surfaces $\tilde M$) to the CV2.0 state complexity proposal. For general $\Sigma_1$ and $\Sigma_2$, with $\Sigma_1$ deep in the IR bulk region and parametrically smaller than $\Sigma_2$, the minimising $\tilde M$ will be a disjoint union of the null boundaries of two WDW patches - one patch each for $\Sigma_1$ and $\Sigma_2$ - and since $\Sigma_1$ is so much smaller than $\Sigma_2$, the volume of $\Sigma_1$'s WDW patch is subleading with respect to the $\Sigma_2$'s WDW patch volume and can be neglected at leading order.
    The takeaway is that while $\Sigma_1$ should be non-empty for our cost proposals, the minimal cost is insensitive to its size and position, as long as $\Sigma_1$ is much smaller than $\Sigma_2$ - this is a reflection of the insensitivity of the leading divergence of state complexity to the initial reference state.

  6. It is correct that in Lorentzian signature we restrict to nowhere-spacelike $\tilde{M}$. In Euclidean signature no such restriction is imposed. In section 4.1, the section in question, as stated in the second sentence, we are considering ``any asymptotically \textbf{Euclidean} AdS spacetime". The (Euclidean) cost proposal in equation (4.2) gives the minimal volume surface $\tilde{M}^\ast$ after optimisation. Upon continuation to Lorentzian signature this a maximum volume surface (with the assumption that this surface is time-reflection symmetric) giving us the $CV$ proposal. The same cost proposal in equation (4.2) directly applied in Lorentzian signature would be invalid.

  7. Our prescription minimises cost over all allowed $\tilde M$. To reach (4.24) we performed the minimisation in two steps: first we minimise for fixed Euclidean time evolution $T$, then we minimise with respect to $T$.

  8. We agree with both referees that section 5 could have been tied together better with the rest of the paper. We have hence extended the paragraph at the beginning of section 5 in order to improve this.

---

## Round 2 · Referee Report · Anonymous (Referee 2) · 2022-8-11

Strengths

1-Intriguing translation of the notion of complexity of preparation of a bulk state on a Cauchy slice to the cost of a the field theoretic causal evolution of this slice.
2-Clearly written with the motivation and logic clearly laid out.
3-Appropriately and amply references the existing literature.
4-Opens up many directions for future inquiry.

Weaknesses

1-Slight technical issues, which are readily fixed.
2-Deriving some complexity proposal from some cost proposal can seem to fly in the face of the rather obvious differences between these two things. The paper is begging the question here. Again, I think this is easily remedied.
3-There are a couple points where a simplifying assumption is made and no further comment is made as regards the more general situation.
4-The figures already in the paper are extremely helpful, but I think it could do with some more.
5-I agree with the first referee that section 5 seems rather disjointed from the rest of the paper than are the other bulk sections.

Report

I believe that the article does meet the acceptance criteria of SciPost Physics. However, my suggestion is that the authors seriously consider and address the points raised by myself and the first referee beforehand. My feeling is that the changes being proposed constitute "minor" rather than "major" revisions.

Numbers refer to corresponding numbers in "Weaknesses" section:
1-I take it that $\Sigma_{1,2}$ are closed, or else that they may as well be since we would really just take their closure anyway in all the arguments and constructions in the paper. So, I just take slight issue with the statement of taking $\Sigma_1$ to be the empty set in Figure 2 and at corresponding points in the main text. I think you really mean a point, as the first referee also says. Once this picture is embedded in AdS, then point 05 of the first referee is relevant.
2-The two differences between complexity and cost that jump out are UV-finiteness of one and not the other and subadditivity versus additivity. It may be obvious, but I think it's worth pointing out that this does not preclude the two being optimized by the same solution. I think this is the sense in which the authors derive some complexity proposals "from" cost proposals. They are certainly not somehow equating complexity with cost, but showing how the proposals can align in certain cases.
3-The first of these is in the middle of Section 4.1 (top of page 27 in current draft) where it is assumed that the "minimal volume surface lies on a time reflection symmetric slice $\Sigma$ of the Euclidean space", implying vanishing extrinsic curvature tensor. In truth, only the trace of the latter must vanish. Could the authors address the more general case? I don't know how much more difficult this is.
The second case is in 4.3 (top of page 30 in current draft). Here it is assumed that whatever corrections would have to be made in order to ensure positivity in "problematic fringe cases" are "negligible when evaluated on pure AdS." I think this is a reasonable assumption and can/should be justified by the fact that there is no negativity of cost problem in pure AdS with matter in its vacuum configuration to begin with. Also, if the putative corrections evaluated on pure AdS weren't negligible (either small or constant or some such condition), then you wouldn't derive CV anymore and it is hard to imagine what would come out instead. Nevertheless, I suppose this is a logical possibility worth pointing out.
4-Again, the figures are already very helpful. However, I think that the discussion in the last two paragraphs of Section 4.2 would benefit from some visualization.
5-I apologize here for not being able to come up with a concrete suggestion here. But I'm sure that the authors have a strong motivation for redirecting their attention to their previous work [35]. It would be nice to make these connections at the beginning of section 5. As far as I can tell, the only connection made explicit is in the middle of page 41 of the current draft when some mention of Section 4.3.2 is made with regards to geodesics. Is this the only concrete connection? In any case, opening the section with it would be helpful for the flow.

Requested changes

The following are just typos or minor language issues that I found and might as well point out:
1-Page 4: "such a bulk subregions" --> "such bulk subregions" (keep it plural; that's consistent with the first part of the sentence).
2-Page 5: "...this paper only considers..." I believe "this" refers to [22], but it can easily be confused with the present article. I think you can just replace "this paper" with "it".
3-Page 8: "The bulk theories we consider path integrals in are..." --> "The bulk theories containing the path integrals which we consider are..." or something like this.
4-Page 11: "it is dual a bulk" --> "it is dual to a bulk"
5-Page 20: "the space possible" --> "the space of possible"
6-Page 23: "such as as minimum" --> "such as minimum"
7-Page 24: "an on-shell solutions" --> "an on-shell solution"; and "it is does" --> "it does"
8-Page 30: The last sentence before Section 4.3.1 is rather awkwardly stated. At minimum, I think commas before and after "or necessarily expect that" are called for.

  • validity: good
  • significance: good
  • originality: high
  • clarity: good
  • formatting: good
  • grammar: excellent

Author:  A. Ramesh Chandra  on 2022-10-06  [id 2890]

(in reply to Report 2 on 2022-08-11)
Category:
answer to question
reply to objection
correction

We thank the referee for the helpful report and for spotting out several typos. In the following we address the points raised by the referee, numbering our replies in correspondence with the referee's points.

  1. It was incorrect to call $\Sigma_1$ the empty set, as we do for example in section 2.3, and we will change where we have written empty set.

  2. We emphasize the point that while cost and complexity have different properties, optimisation of a cost could still lead to a complexity.

  3. For time reflection-symmetric surfaces, the full extrinsic curvature tensor is zero, not just its trace. As we state in the same paragraph, this assumption was made so that we can relate the second order shape variations in both the signatures. They flip signs when we continue to Lorentzian signature only when the extrinsic curvature tensor is zero. In the more general case, Euclidean minimal volume surfaces aren't related to Lorentzian maximal volume surfaces.

  4. We added a new figure in section 4.2 explaining the derivation of CV2.0 from cost equals bulk volume.

  5. We agree with both referees that section 5 could have been tied together better with the rest of the paper. We have hence extended the paragraph at the beginning of section 5 in order to improve this.

---

## Round 3 · Referee Report · Anonymous · 2022-10-17

Report

The authors have addressed all of my concerns well. I wholeheartedly recommend the article be accepted for publication.

---

## Round 3 · Referee Report · Anonymous · 2022-10-17

Report

I think the authors have addressed most of my comments and questions in their replies.

My last question is related to the previous point 5 and the new figure 17 in section 4.2.
I thought before that the timelike surface $\tilde{M}$ is defined by connecting $\partial \Sigma_1$ and $\partial \Sigma_2$.
From the new figure 17 and the description around eq.(4.12). I guess my previous
understanding is wrong. As shown in figure 17, the timelike surface $\tilde{M}$ (with respect to which the optimization is performed) could even be piecewise or discontinuous.
If this is correct, I guess it is better to stress this point at the beginning of the paper.

And I'm happy to recommend the paper for publication.

  • validity: good
  • significance: good
  • originality: high
  • clarity: good
  • formatting: good
  • grammar: excellent

Author:  A. Ramesh Chandra  on 2022-11-03  [id 2981]

(in reply to Report 1 on 2022-10-17)
Category:
answer to question

Thank you for the above question. It is not incorrect to think of $\tilde{M}$ as a surface connecting $\partial\Sigma_1$ and $\partial\Sigma_2$. More importantly, as we optimise a given cost, we keep $\partial\tilde{M} = \partial\Sigma_1\cup\partial\Sigma_2$ fixed. This condition is still satisfied in the case illustrated in new figure 7. Here, the minimising (timelike) $\tilde{M}$ could be a disjoint union depending on the location of $\Sigma_1$. In the limit as $\Sigma_1$ is taken to a point, the contribution from smaller component of $\tilde{M}$ shrinks to zero, and we remain with a connected $\tilde{M}$.

---

## Round 3 · List of Changes

1. Added a new figure (figure 7) in section 4.2, to explain the discussion of obtaining CV2.0 from cost=bulk volume.
2. Expanded the paragraph at the beginning of section 5 to connect it better with the rest of the paper.
3. The highly coarse-grained initial state $\Sigma_1$ is changed to a point, instead of being an empty set.
4. Minor corrections: fixed typos and rephrased slightly confusing statements.

---

## Editorial Decision

published